# Effect of intrapulmonary percussive ventilation on intensive care unit length of stay, the incidence of pneumonia and gas exchange in critically ill patients: A systematic review

Anwar Hassan [1,2]*, William Lai[1], Jennifer Alison[2], Stephen Huang[1,2], Maree Milross[2]

**1** Nepean Hospital, Nepean Blue Mountains Local Health District, Penrith, NSW, Australia, **2** Sydney School of Health Sciences, Faculty of Medicine and Health, The University of Sydney, Camperdown, NSW, Australia

* anwarpt77@yahoo.com.au

**Data Availability Statement:** The search results and results of data extraction for this systematic

## Abstract

### Background

Pulmonary complications such as pneumonia, pulmonary atelectasis, and subsequent respiratory failure leading to ventilatory support are a common occurrence in critically ill patients. Intrapulmonary percussive ventilation (IPV) is used to improve gas exchange and promote airway clearance in these patients. The current evidence regarding the effectiveness of intrapulmonary percussive ventilation in critical care settings remains unclear. This systematic review aims to summarise the evidence of the effectiveness of intrapulmonary percussive ventilation on intensive care unit length of stay (ICU-LOS) and respiratory outcomes in critically ill patients.

### Research question

In critically ill patients, is intrapulmonary percussive ventilation effective in improving respiratory outcomes and reducing intensive care unit length of stay.

### Methods

A systematic search of intrapulmonary percussive ventilation in intensive care unit (ICU) was performed on five databases from 1979 to 2021. Studies were considered for inclusion if they evaluated the effectiveness of IPV in patients aged ≥16 years receiving invasive or non-invasive ventilation or breathing spontaneously in critical care or high dependency units. Study titles and abstracts were screened, followed by data extraction by a full-text review. Due to a small number of studies and observed heterogeneities in the study methodology and patient population, a meta-analysis could not be included in this review. Outcomes of interest were summarised narratively.

review are included in the Supporting Information files.

**Funding:** The authors received no specific funding for this work.

**Competing interests:** The authors have declared that no competing interests exist.

**Abbreviations:** COPD, chronic obstructive pulmonary disease; CPT, chest physiotherapy; $FiO_2$, fraction of inspired oxygen; IPV, intrapulmonary percussive ventilation; ICU-LOS, intensive care unit length of stay; NIV, non-invasive ventilation; $PaCO_2$, partial pressure of arterial carbon dioxide; $PaO_2$, partial pressure of arterial oxygen; $SpO_2$, saturation of peripheral oxygen.

## Results

Out of 306 identified abstracts, seven studies (630 patients) met the eligibility criteria. Results of the included studies provide weak evidence to support the effectiveness of intra-pulmonary percussive ventilation in reducing ICU-LOS, improving gas exchange, and reducing respiratory rate.

## Interpretation

Based on the findings of this review, the evidence to support the role of IPV in reducing ICU-LOS, improving gas exchange, and reducing respiratory rate is weak. The therapeutic value of IPV in airway clearance, preventing pneumonia, and treating pulmonary atelectasis requires further investigation.

## Introduction

The incidence of pulmonary complications such as pulmonary atelectasis, pneumonia (including ventilator-associated pneumonia), and acute respiratory failure is high in critical care patients [1, 2]. The incidence of ventilator-associated pneumonia can be as high as 27% amongst mechanically ventilated patients [3]. Studies have shown that 16% of critically ill patients have been reported to develop acute respiratory failure, which is associated with prolonged intensive care unit stay, resulting in significantly higher mortality than non-respiratory failure patients [2, 4–7]. Increased morbidity and mortality contribute to the burden on the health care system and lead to poor health-related outcomes [6–8]. Multi-modal physiotherapy plays a role in the management of these critically ill patients [9]. Chest physiotherapy (CPT) interventions such as chest percussion & vibrations, postural drain-age, positioning, thoracic expansion exercises, manual hyperinflation, ventilator hyperinfla-tion, and airway suctioning aim to promote airway secretion clearance, increase alveolar recruitment, minimise pulmonary shunting, and optimise ventilation and perfusion (V/Q) matching [10, 11]. In addition to these CPT interventions, intrapulmonary percussive venti-lation (IPV) is used in patients with underlying pulmonary atelectasis, excessive airway secretions, and respiratory failure [12–15].

IPV is a non-continuous form of high-frequency ventilation delivered by a pneumatic device that provides small bursts of sub-physiological tidal breaths at a frequency of 60–600 cycles/minute superimposed on a patient's breathing cycle [16–18]. The high-frequency breaths create shear forces causing dislodgement of the airway secretions. Furthermore, the IPV breath cycle has an asymmetrical flow pattern characterised by larger expiratory flow rates, which may propel the airway secretions towards the central airway [18]. In addition, the applied positive pressure recruits the lung units, resulting in a more homogeneous distribution of ventilation and improved gas exchange [19]. In acute care and critical care settings, IPV intervention is used in a range of patients, from spontaneously breathing patients to those receiving invasive mechanical ventilation where IPV breaths can be superimposed on a patient's breathing cycle or superimposed on breaths delivered by a mechanical ventilator. The most common indications for IPV use are reported as removal of excessive bronchial secre-tions, improving gas exchange, and recruitment of atelectatic lung segments [12–14, 18]. In the last two decades, studies have reported IPV in the critical care setting to be effective in improving outcomes in patients with an acute exacerbation of chronic obstructive pulmonary

disease (COPD), burns, pulmonary atelectasis, and those with post-abdominal or thoracic surgery [14, 15, 20, 21]. Despite the available studies, the overall evidence regarding its effectiveness in critical care settings remains unclear. Recently, Reychler and colleagues (2018) [22] summarised the effectiveness of IPV in promoting airway clearance and gas exchange in chronic lung diseases such as COPD, cystic fibrosis, and bronchiectasis. The question regarding the role of IPV in preventing or reversing atelectasis and reducing the incidence of pneumonia in critically ill patients remains unanswered. Most of the studies reviewed by Reychler and colleagues (2018) included stable patients; hence the findings of their review are not applicable to critically ill patients [22]. The objective of this systematic review was to summarise the evidence for the effectiveness of IPV in improving outcomes such as intensive care unit length of stay (ICU-LOS), gas exchange, respiratory rate, the incidence of pneumonia, and reversing or preventing atelectasis in critical care patients.

## Methods

This systematic review followed recommendations from the Preferred Reporting Items for Systematic Reviews and Meta-Analysis (PRISMA) [23]. The review protocol was registered in the International Prospective Register of Systematic Reviews (PROSPERO) before conducting the database search (Registration ID: CRD42018115517).

### Search strategy

A systematic search of the literature was conducted in two stages. The first stage included database searches on MEDLINE, EMBASE, CINAHL, Web of Science, and PEDro, from 1979 (when IPV was first introduced) to February 2021. The details of the search strategy and keywords used are presented in S1 Table. The second stage included searching the relevant clinical trial registries (including ClinicalTrials.gov, ANZCTR, WHO, EUCTR, ATS, PROSPERO). A manual search of the reference lists of the included studies was also conducted.

### Inclusion and exclusion criteria

Studies were considered for inclusion if they evaluated the effectiveness of IPV in patients aged ≥16 years receiving invasive or non-invasive ventilation (NIV) or breathing spontaneously in critical care units for acute or acute on chronic impairment of respiratory function. Studies that included stable patients in the inpatient, outpatient, or community-based settings were excluded.

Due to the limited number of studies, randomised controlled trials (RCT), quasi-randomised trials, randomised crossover studies, observational studies, comparative studies, experimental designs with random allocation, and retrospective studies were all considered. Studies that reported the effects of IPV, high-frequency ventilation, and high-frequency oscillation where these interventions were primarily used intermittently for a short duration to promote airway clearance, reverse, or treat pulmonary atelectasis, or to improve gas exchange were included, whereas the studies that used these interventions to provide continuous mechanical ventilation were excluded.

Studies that measured ICU-LOS and examined the physiological variables such as changes in the saturation of peripheral oxygen ($SpO_2$) and partial pressure of arterial oxygen ($PaO_2$), partial pressure of arterial carbon dioxide ($PaCO_2$) measured by arterial blood gas analysis and airway clearance were included. Other reported outcomes, such as pulmonary atelectasis and respiratory rate, were also included.

## Study review and data extraction

Duplicates were removed using Covidence® software, followed by a manual search for duplicates. The remaining articles were screened independently by two authors (AH and WL) by reviewing the study titles and abstracts (inter-rater reliability, Kappa 0.84). Authors of the eligible studies with published abstracts only or papers with missing or insufficient data were contacted via email for the full-text article or raw data; the study was excluded if no response was obtained within four weeks. A full-text review was conducted by two authors (AH and WL), and ineligible studies were excluded (Fig 1). In case of disagreement, a third reviewer (MM) independently reviewed the study. Included studies were then used for data extraction and quality assessment for the risk of bias.

## Assessment of quality and risk of bias

The quality and risk of bias for the studies that used random allocation were assessed according to the Cochrane Collaboration assessment tool during the data extraction phase using the Covidence® software [24]. For the included literature, the following risk of bias

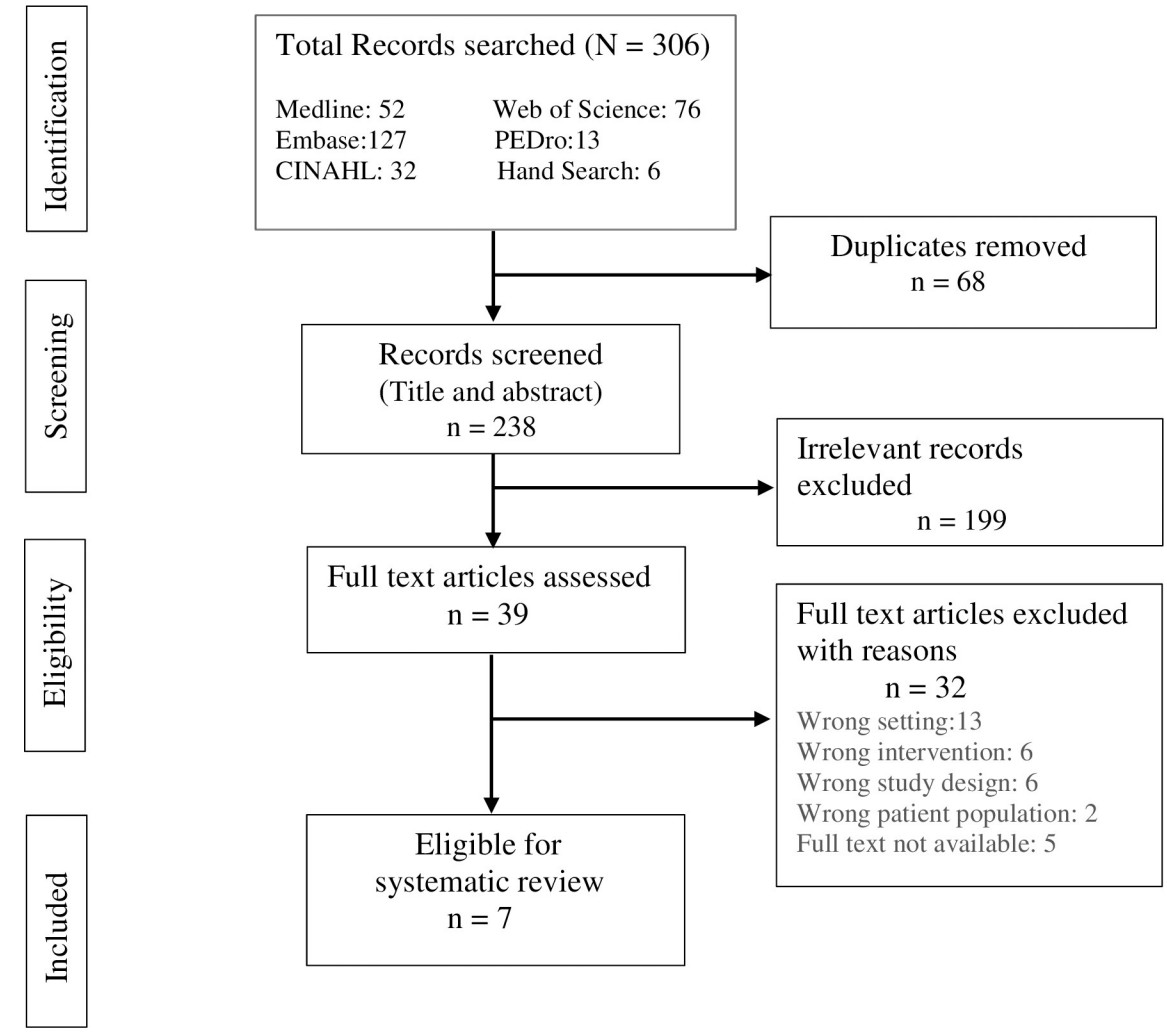

**Fig 1. PRISMA flow chart.**

domains was assessed: random sequence generation, allocation sequence concealment, blinding of participants, blinding of the therapist, blinding of outcome assessors, incomplete outcome data, selective outcome reporting, and overall risk of bias (Table 1). The level of risk of bias was assessed under three categories: 1) high, 2) low, and 3) unclear. In addition, the Physiotherapy Evidence Database (PEDro) scale was also used to assess and summarise the study quality for all the included studies (Table 2) [25]. The PEDro scale allows assessment on ten different domains to determine the study quality. Based on the total PEDro score, studies can be categorised into; "poor" (score 0–3), "fair" (score 4–5), "good" (score 6–8), and "excellent" (score 9–10). Overall, based on the Cochrane assessment tool, three out of four studies appear to have a low risk of bias whereas, in one study [26], the risk of bias was high. On the PEDro scale, the quality of the studies ranged from "poor" to "good."

## Outcome measures

The primary outcome of interest was ICU-LOS. Secondary outcomes included $PaO_2$, the ratio of the partial pressure of arterial oxygen and fraction of inspired oxygen ($PaO_2/FiO_2$), $PaCO_2$, airway clearance, the incidence of pneumonia, respiratory rate, and pulmonary atelectasis.

Due to the small number of studies and observed heterogeneities in the study methodology and patient population, all the outcomes were summarised narratively.

# Results

The database search yielded a total of 306 studies. After removing 68 duplicates, the titles and abstracts were screened for 238 studies. After excluding 199 irrelevant studies, the remaining 39 full-text articles were assessed for their eligibility. The reviewers identified seven studies including 630 patients, which met the eligibility criteria for systematic review (Fig 1). The study characteristics are described in Table 3.

## Study characteristics

Among the included studies, four studies were RCTs, one was a quasi-RCT with a historical control group, and the remaining two were prospective observational studies. The observational studies did not assign a control group for comparison [14, 27]. Study sample sizes ranged from 10 patients to 419 patients [14, 21] (Table 3). All the studies were conducted in the critical care setting, which included patients who were mechanically ventilated (34%), post-extubation (6%), requiring NIV (18%), and the remaining (42%) were requiring high oxygen therapy ($FiO_2 \geq 40\%$) and continuous positive airway pressure support.

**Table 1. Cochrane assessment of the risk of bias.**

| Study | Random sequence generation | Allocation concealment | Blinding of participants and personnel | Blinding of outcome assessors | Incomplete outcome data | Selective reporting | Other bias |
|---|---|---|---|---|---|---|---|
| Antonaglia et al. (2006) [13] | Low | Low | High | High | Low | Low | Low |
| Vargas et al. (2005) [15] | Low | Low | High | High | Unclear | Unclear | Low |
| Clini et al. (2006) [12] | Low | Low | High | Low | Low | Low | Low |
| Dimassi et al. (2011) [26] | Low | High | High | Unclear | Low | High | High |

**Table 2. Quality assessment using PEDro scale [25].**

| Study | Eligibility criteria | Random allocation | Concealed allocation | Groups similar at baseline | Subject blinding | Therapist blinding | Assessor blinding | < 15% dropouts | Intention to treat analysis | Statistical comparison in groups | Measure of variability | Total score |
|---|---|---|---|---|---|---|---|---|---|---|---|---|
| **Antonaglia 2006 [13]** | Y | Y | Y | Y | N | N | N | N | N | Y | Y | 5/10 |
| **Vargas 2005 [15]** | Y | Y | Y | Y | N | N | N | Y | N | Y | Y | 6/10 |
| **Huynh 2019 [21]** | Y | N | N | N | N | N | N | Y | Y | Y | Y | 4/10 |
| **Clini 2006 [12]** | Y | Y | Y | Y | N | N | Y | Y | N | Y | Y | 7/10 |
| **Dimassi 2011 [26]** | Y | Y | N | Y | N | N | N | Y | N | Y | Y | 5/10 |
| **Vargas 2009 [27]** | Y | N | N | N | N | N | N | Y | N | N | Y | 2/10 |
| **Tsuruta 2006 [14]** | Y | N | N | N | Y | N | N | Y | N | N | Y | 3/10 |

Y = Yes, N = No. PEDro score (0–3 = poor, 4–5 = fair, 6–8 = good, 9–10 = excellent)

## Patient characteristics

Patient's ages ranged from 18 to 95 years (mean age 63.7 years), with 60% males. One study did not report the mean age [26]. Common clinical conditions were post thoracic, aortic, or abdominal surgeries [21], acute respiratory failure secondary to COPD [13, 15, 27], post-extubation respiratory failure [26], pulmonary atelectasis [14], and tracheostomised patients with impaired respiratory function [12] (Table 3).

## Intervention

In the included studies, the most common indications to use IPV interventions were to improve gas exchange, promote airway clearance and prevent or reverse pulmonary atelectasis. The treatment application varied among the studies. In most studies, IPV was delivered via a facemask or mouthpiece, whereas for those who were mechanically ventilated, IPV was delivered via an in-line ventilator circuit [14, 21] (Percussionaire Corporation, Sandpoint, ID, USA), and Metaneb$^®$ (Hill-ROM corporation, USA). All these devices work on the same mechanical principles and use phasitron to deliver similar breath frequency (200 to 300 cycles/minute) and airway pressures (Table 3). The treatment dosage, such as duration and frequency of sessions per day, varied across the studies. For instance, the duration of a single treatment session ranged from 10 to 30 minutes, and the number of sessions ranged from a single session a day to up to six sessions a day [21]. The frequency of delivered breaths remained between 200 to 300 cycles per minute in all the included studies, whereas the airway pressure varied from 5 to 35 cmH$_2$O (Table 3). Notably, most of the studies did not specify the patient's position during the treatment. IPV intervention was compared to CPT [12, 13, 21] which was reported as being used to promote airway clearance, improve gas exchange, and increase or restore lung volume. CPT included chest clapping, postural drainage, expiration with open glottis, incentive spirometer and mobilisation (Table 3). Duration of CPT session ranged from 30 minutes to 60 minutes once or twice a day. In two studies [15, 26], the control group received standard medical treatment, which included oxygen therapy, non-invasive ventilation, sitting up in bed (45 degrees), nebulised bronchodilators, and corticosteroids (Table 3).

## Outcome measures

The common outcomes reported were ICU-LOS, the incidence of pneumonia, changes in PaO$_2$, PaO$_2$/FiO$_2$, PaCO$_2$, and respiratory rate. Less commonly, studies also recorded the incidence of pulmonary atelectasis, changes in diaphragmatic work, duration of mechanical

**Table 3. Study summary.**

| Author (Year) | Study design, N | Population | Inclusion and exclusion criteria | Intervention(s) | Outcomes |
|---|---|---|---|---|---|
| Antonaglia et al. (2006) [13] | RCT | Admitted to ICU with an acute exacerbation of COPD | **Inclusion** | **IPV group** | **Gas exchange** |
| | N = 80 | | Admitted to ICU within 12 hrs of emergency department admission | Duration: 25 to 30 min (via mouthpiece) twice / day | After the 1st IPV session: |
| | | | | IPV Setting: 225 cycles / min, $P_{AW} < 40$ cmH$_2$O | IPV: $\uparrow$ pH**, $\downarrow$ PaCO$_2$**, $\uparrow$ PaO$_2$/FiO$_2$** |
| | IPV: 20 | Age: 69.75±6.56 years | RR >25 breaths/min | | At discharge: |
| | | Gender: Not reported | PaCO$_2$ >50 mmHg | **Control group** | IPV: pH (NS), $\downarrow$ PaCO$_2$**, $\uparrow$ PaO$_2$/FiO$_2$** |
| | | | pH: 7.10–7.35 | Standard medical care (NIV via facial mask) | Control: pH (NS), $\downarrow$ PaCO$_2$**, $\uparrow$ PaO$_2$/FiO$_2$** |
| | CPT: 20 | | | | **Cardio-respiratory parameters** |
| | | | | **CPT group** | After the 1st session: |
| | Control: 40 | | **Exclusion** | Duration: 25 to 30 min CPT (chest clapping, mobilisation, and postural drainage, and expiration with the glottis open) twice / day | IPV: $\downarrow$ RR**, $\downarrow$ HR* |
| | | | Need for emergency intubation | | **ICU-LOS** (days) |
| | | | GCS < 8 | | IPV: 7 [6–8], Control:10 [9–11]**, CPT: 9 [7.7–9.5] ** |
| | | | Haemodynamic instability | | **Pneumonia** |
| | | | Failure > two additional organs | | IPV: 2, Control: 11(NS), CPT: 4(NS) |
| Vargas et al. (2005) [15] | RCT | Admitted to ICU with an acute exacerbation of COPD | **Inclusion** | **IPV group** | **Gas exchange** |
| | | | | | IPV: $\uparrow$ PaO$_2$*, $\downarrow$ PaCO$_2$* |
| | N = 33 | | Admitted to ICU as an emergency with acute exacerbation of COPD | Duration: 30 min (via a face mask) twice / day | Control: Not reported |
| | | | | IPV Setting: 80–650 cycles / min, $P_{AW}$: 5-35cmH$_2$O, I/E: 1/2.5. | **Cardio-respiratory parameters** |
| | | | | | IPV: $\downarrow$ RR*, Control: Not reported |
| | IPV: 16 | Age = 69.71±5.44 years | RR $\geq$ 25 breaths/min | IPV sessions were stopped when a RR of < 25/min and a pH > 7.38 was reached | **ICU-LOS** (days) |
| | | | PaCO$_2$ > 45 mmHg | | IPV: 6.8±1.0, Control: 7.9±1.3* |
| | Control: 17 | Gender: Not reported | pH: 7.35 to 7.38 on room air > 10 minutes | Same drug protocol as the control | **Need for NIV** |
| | | | | | IPV: (0) 0%, Control: (6) 35.3%* |
| | | | | **Control group** | |
| | | | **Exclusion** | Standard medical care | |
| | | | Need for emergency intubation | Supplemental oxygen to maintain SpO$_2$ of 88–92% | |
| | | | GCS $\leq$ 8 | HOB elevated at a 45-degree angle | |
| | | | Haemodynamic instability | Drug protocol including nebulised bronchodilators and corticosteroids | |
| | | | Failure > two additional organs | | |
| | | | Tracheostomy | | |
| | | | Pneumothorax | | |
| | | | Recent oral/oesophageal/gastric surgery | | |

*(Continued)*

**Table 3.** (Continued)

| Author (Year) | Study design, N | Population | Inclusion and exclusion criteria | Intervention(s) | Outcomes |
|---|---|---|---|---|---|
| Dimassi et al. (2011) [26] | Randomised (crossover) to receive IPV or NIV | ICU patients at risk of post-extubation failure | **Inclusion** | **IPV group** | **Gas exchange** |
| | | | Intubation > 48 hours, who tolerated SBT plus at least 2 of the following: | Duration: 20 min (via a face mask) | IPV: $PaCO_2$ (NS), $PaO_2/FiO_2$ (NS) |
| | | Age: 73 [58–75] years | | IPV Setting: 250 cycles / min, driving pressure: 1.2 bar, I/E: 1 / 2.5 | Control: ↓$PaCO_2$**, $PaO_2/FiO_2$ (NS) |
| | | | | | **Cardio-respiratory parameters** |
| | | | | | IPV: ↓RR**, Control: ↓RR** |
| | | Age > 65 years | | **Control group** | **Diaphragmatic work (PTPdi/ breath)** |
| | | Gender: M 14, F 3 | Underlying heart or respiratory failure, and | NIV | IPV: ↓20%, Control: ↓35% (NS) |
| | N = 17 | | APACHE II score > 12 | NIV Settings: Delivered via ventilator in pressure-support mode with PEEP. Tidal volume target 6-8ml/kg, PEEP 4–5 $cmH_2O$ | |
| | | | **Exclusion** | | |
| | IPV: 8 | | Tracheostomy | | |
| | NIV: 9 | | Facial or cranial trauma or surgery | | |
| | | | Recent gastric/oesophageal surgery | | |
| | | | Active UGI bleeding | | |
| | | | Lack of cooperation | | |
| | | | Limit of therapy in ICU | | |
| Clini et al. (2006) [12] | RCT | Tracheostomised patients randomised to two treatment groups | **Inclusion** | **IPV group** | **Gas exchange** |
| | | | Mechanically ventilated ≥ 14 days | IPV + CPT | IPV group vs. Control group during the treatment period: ↓ pH (NS), ↑ $PaO_2$ (NS), ↑ $PaO_2/FiO_2$*, |
| | | | Passed the SBT for at least 72 hours | Duration: 10 min (via tracheostomy tube) twice / day | |
| | | | | | ↓ $PaCO_2$ (NS) |
| | N = 46 | | Stable, conscious and able to adhere to active physiotherapy treatment | IPV Setting: 200–300 cycles / min | **Cardiorespiratory parameters** |
| | | Age: 68.96±9.06 years | Sputum > 40ml/day | $P_{AW}$: 40 cm $H_2O$, I/E: 1:1.2 | IPV group vs. Control group during the treatment period: ↑ MEP* |
| | IPV: 24 | Gender: M 28, F 18 | **Exclusion** | **Control group** | **Pneumonia** |
| | CPT: 22 | | | CPT for one hour, twice / day | Day 5: IPV: 3, CPT: 5* |
| | | | Persistent alterations of the sensorium | | One month: IPV: 0, CPT:2* |
| | | | Haemodynamic/respiratory instability | | |
| | | | Reconnection to ventilator < 72 hours | | |
| | | | On continuous sedatives and vasopressors | | |
| Huynh et al. (2019) [21] | Multicentre prospective observational study | Post-thoracic, upper abdominal and aortic surgery patients admitted to ICU | **Inclusion** | **IPV group** | **Gas exchange** Not measured |
| | | | Age ≥ 18 years post thoracic, upper abdominal and aortic surgery in addition to ICU | Received IPV in addition to standard care | **Cardiorespiratory parameters** Not measured |
| | | | | Duration: 10 min per session | **ICU-LOS** (days) |
| | | | ASA class ≥ 3 OR 1 and 2 with one or more of the following: current smoker, COPD, BMI ≥ 30, age > 75 years | IPV for intubated patients six times / day | IPV: 3.4±3.5, CPT 5.4±8.7 (NS) |
| | | | | non-intubated patients 4 times / day | **Time on mechanical ventilation** (hours) |
| | N = 419 | Age: 59.25±14.73 years | | IPV setting: 170–230 cycles / min | IPV: 29.7±44.8 to 94.1±199.2* |
| | | Gender: M 246, F 173 | | | **Pneumonia:** 3 (1.4%) in both groups |
| | IPV: 209 | | **Exclusion** | **Control group** | **Hospital LOS** (days) |
| | CPT: 210 | | Contraindication to positive pressure therapy; untreated tension pneumothorax, organ transplant, spinal surgery, and positive pressure ventilation at baseline | Standard care including incentive spirometer Additional respiratory treatment was provided based on clinical indication | IPV: 6.78±4.5, Control: 8.4 ±7.9* |

*(Continued)*

**Table 3.** (Continued)

| Author (Year) | Study design, N | Population | Inclusion and exclusion criteria | Intervention(s) | Outcomes |
|---|---|---|---|---|---|
| Tsuruta et al. (2006) [14] | Prospective observational study | Mechanically ventilated patients with compression atelectasis | **Inclusion** | **IPV group** | **Gas exchange** |
| | | | Acute respiratory failure due to compression atelectasis unresolved by conventional mechanical ventilation | IPV delivered via in-line ventilator circuit | Pre IPV compared to 24h post IPV |
| | | | | Duration and frequency: Not reported | $PaO_2/FiO_2$ 189±63 to 280±55** |
| | | | | Setting: 300 cycles / min | $PaCO_2$: 38 to 37 (NS) |
| | N = 10 | Age: 52±19 years | | | **Cardiorespiratory parameters** |
| | | | BMI > 25 | | HR (NS) |
| | | | | | **Improvement of atelectasis** |
| | | | | | Seven improved on chest CT scans |
| | IPV: 10 | Gender: M 8, F 2 | **Exclusion** | | Ten improved on chest radiographs |
| | Control: not assigned | BMI: 31±6 | Infiltrations induced by infection and drugs | | |
| Vargas et al. (2009) [27] | Prospective observational study | COPD patients with expiratory flow limitation were screened following extubation | **Inclusion** | **IPV group** | **Gas exchange** |
| | | | Diagnosis of COPD deemed stable 1 hour after extubation with: | Duration: 30 min (via a full-face mask) | ↑ pH*, ↑ $PaO_2$*, ↓ $PaCO_2$*, ↑ $SpO_2$* |
| | | | | | **Cardio-respiratory parameters** |
| | | | | | HR (NS), ↓ RR* |
| | | | | IPV Setting: 250 cycles / min | Expiratory flow limitation: ↓ 31%* |
| | | | RR < 30/min | $P_{AW}$: 20 cmH$_2$O, I/E: 1/2.5 | Airway occlusion pressure: ↓ 28%* |
| | | | Lack of respiratory acidosis with a pH > 7.35 | Supplemental oxygen was interfaced into the mask to maintain $SpO_2$ 88–92% | |
| | N = 25 | Age: 63±8 years | **Exclusion** | | |
| | | Gender: M 15, F 10 | Need for emergency intubation | | |
| | | | GCS ≤ 8 | | |
| | IPV: 25 | BMI: 26±3 | Hemodynamic instability | | |
| | Control: not assigned | | Failure > two additional organs | | |
| | | | Tracheostomy | | |
| | | | Pneumothorax | | |
| | | | Recent oral/oesophageal/gastric surgery or facial deformity | | |

RCT = randomised controlled trial, IPV = intrapulmonary percussive ventilation, COPD = chronic obstructive pulmonary disease, CPT = chest physiotherapy, SBT = spontaneous breathing trial, RR = respiratory rate, HR = heart rate, MAP = mean arterial pressure, $P_{AW}$ = airway pressure, MEP = maximal expiratory pressure, HOB = head of bed, I:E = inspiratory to expiratory cycle ratio, APACHE II = acute physiology and chronic health evaluation II, PPC = postoperative pulmonary complication, UGI = upper gastrointestinal, NIV = non-invasive ventilation, PTPdi = diaphragmatic pressure-time product, $V_T$ = tidal volume, MEP = maximal expiratory pressure, ASA = American society of Anaesthesiologists, BMI = body mass index, NS = not significant,

* = statistically significant $p \leq 0.05$,

** = statistically significant $p \leq 0.01$

ventilation, need for NIV, hospital length of stay, and mortality. Some of these physiological outcomes ($PaO_2$, $PaO_2/FiO_2$, $PaCO_2$ and respiratory rate) were recorded daily before and after the intervention and also at the time of discharge, whereas one study recorded these outcomes at five-day intervals for up to 15 days [12].

**ICU length of stay.** Among the included studies, three studies reported on ICU-LOS [13, 15, 21]. Antonaglia and colleagues (2006) [13] randomly allocated 40 critically ill patients with

an acute exacerbation of COPD to the IPV group (n = 20) or CPT group (n = 20), where patients in both the groups were treated with NIV. In addition to NIV, patients in the CPT group received standard chest physiotherapy for 25–30 minutes, and those in the intervention group received 25–30 minutes of IPV twice a day (Table 3). Antonaglia et al. (2006) also included a historical control group (n = 40) for comparison that received standard medical treatment. A significantly shorter ICU-LOS in the IPV group (median = 7 [6, 8] days) than the control group was reported (median = 10 [9, 11] days), median difference -2.0 days (95% CI: -2.19; -1.81 days, $p < 0.01$). Similarly, a multicentre study by Huynh et al. (2019) [21] evaluated the effect of IPV in 419 postoperative (upper abdominal, aortic, and thoracic surgery) patients admitted to ICU, where the intervention group (n = 209) received IPV for 10 minutes four to six times a day, and the historical control group (n = 210) received CPT mainly in the form of incentive spirometry. Out of 419 patients, the ICU-LOS was reported only for 161 patients (Intervention = 79, Control = 82) where the ICU-LOS was found to be shorter but not statistically significant in the IPV group (IPV: mean 3.3 [3.5] days vs. Control: mean 5.4 [8.7] days, NS). Another study by Vargas et al. (2005) investigated the effects of IPV intervention in 33 patients with COPD with acute respiratory failure where the intervention group (n = 17) received IPV for 30 minutes twice a day, and the control group (n = 16) received standard medical treatment (Table 3). The study reported a significant reduction in length of stay in the IPV group compared to the control group (IPV: mean 6.8 [1.0] days vs. Control: 7.9 [1.3] days, $p < 0.05$) [15].

**Incidence of pneumonia.** Among the included studies, the incidence of pneumonia was reported by three studies [12, 13, 21]. Antonaglia et al. (2006) reported a small difference in the incidence of pneumonia in 40 patients with acute exacerbation of COPD (IPV: 2 vs. CPT: 4, NS) [13]. Similarly, Huynh et al. (2019) did not find any difference in the incidence of pneumonia in 419 patients with upper abdominal and thoracic surgery patients (IPV: 3 vs. CPT: 3, NS) [21]. However, one study reported a significant reduction in the incidence of pneumonia in tracheostomised patients treated with IPV (IPV: 3 vs. CPT: 5, $p < 0.05$) [12].

## Gas exchange

*a) $PaO_2$ and $PaO_2/ FiO_2$ ratio*. Six studies (n = 211) reported an increase in oxygenation in the IPV group [12–15, 26, 27]. The increase in oxygenation was recorded as a change from baseline to post-intervention, as $PaO_2$ and or the $PaO_2/ FiO_2$ ratio.

Antonaglia et al. (2006) reported a significant change in $PaO_2/ FiO_2$ ratio from admission to discharge (seven days) in patients with COPD admitted to ICU (IPV: 173 [27] to 274 [15], Control: 181 [29] to 237 [20], $p < 0.01$) [13]. Similarly, Clini et al. (2006) also reported an improvement in $PaO_2/FiO_2$ ratio in 46 patients with tracheostomy randomised to receive either CPT or IPV in addition to CPT, after 15 days of intervention (IPV: 238 [51] to 289 [52], Control: 240 [34] to 255 [38], $p < 0.05$), median difference 21.65 (95% CI: -11.75 to– 55.05, $p < 0.038$) [12]. Similarly, Vargas et al. (2005) found a significant increase in $PaO_2$ in patients with COPD who received IPV intervention (56.9 [3] to 61 [0.8] mmHg, $p < 0.05$) [15]. These findings of increased oxygenation ($PaO_2/FiO_2$ ratio and $PaO_2$) in the IPV group were consistent with the findings of two observational studies [14, 27]. In contrast, one small study of 17 post-extubation patients, who received IPV intervention and NIV in random order, reported no significant change in the $PaO_2/FiO_2$ ratio [26].

*b) Change in $PaCO_2$*. A total of six studies evaluated a change in the $PaCO_2$ levels [12–15, 26, 27]. Antonaglia et al. (2006) recorded a significant reduction in $PaCO_2$ levels in patients with COPD in IPV and CPT group (IPV: 79 [7] to 58 [5.4], Control: 80 [6.5] to 64 [5.2] mmHg, $p < 0.01$) [13]. Similar findings were reported by Vargas et al. (2005), where a

significant reduction in the $PaCO_2$ levels was seen in the IPV group (IPV: 57.6 [4.5] to 53.5 [2.3] mmHg, $p < 0.05$) [15]. The study did not report any data for the control group for comparison. In another small study by Vargas and colleagues (2009) in 25 patients (with no control group) with acute exacerbation of COPD found a reduction in $PaCO_2$ (IPV: 55.1 [3.7] to 52.5 [2.2] mmHg, $p < 0.05$) [27]. A small (not significant) reduction in $PaCO_2$ levels was also reported by Clini et al. (2006) in the IPV group without any change in the control group [12].

**Respiratory rate.** Among the included studies, four studies evaluated the effects of IPV intervention on a patient's respiratory rate (RR) [13, 15, 26, 27]. Vargas and colleagues (2005) found a significant reduction in RR in COPD patients in the IPV group (36 [2] to 31 [2] breaths per minute, $p < 0.05$) with no change in the control group [15]. In another study in 25 COPD patients, Vargas et al. (2009) reported a small reduction in RR (IPV: 22.6 [2.3] to 21.4 [1.7] breaths per minute, $p < 0.05$) [27]. Similarly, a small but significant reduction in respiratory rate was observed by Dimassi et al. (2011) in 17 post-extubation patients (23, [19–27] to 22, [17–24] breaths per minute, $p < 0.01$) [26]. In contrast, reports from Antonaglia et al. (2006) study did not demonstrate any significant change in RR [13]. Overall, three out of four studies reported a small but significant reduction in respiratory rate post IPV intervention. This small change in RR does not seem to be clinically relevant.

**Airway clearance.** Three studies in this review reported an observed increase in airway clearance with IPV intervention; however, none of them measured the expectorated sputum weight (wet or dry) [12, 13, 15] or other measures of mucous clearance.

**Adverse events and tolerance.** None of the studies reported any major adverse events related to IPV intervention [12–14, 21, 26]. Vargas et al. (2005) reported a single incidence of haemoptysis in one patient, unrelated to IPV intervention [15]. One study did not report on adverse events [27]. Four studies [12, 13, 15, 26], based on their observational findings, stated that the IPV intervention was well-tolerated; none of the studies asked specific questions pertaining to IPV tolerance. A recent multicentre study found one minor episode of IPV intolerance, which resolved quickly, and therapy was resumed 8 hours later [21].

## Discussion

This systematic review synthesised the evidence of the effectiveness of IPV intervention in critical care patients. The findings of this review provide weak evidence to support the effectiveness of IPV intervention in reducing ICU-LOS, improving gas exchange, and reducing the respiratory rate in critically ill patients compared to chest physiotherapy techniques or standard medical management. Similar findings in patients with chronic lung disease have been reported in another systematic review, including 12 studies (278 patients) in patients with acute exacerbation of COPD, cystic fibrosis and bronchiectasis in a range of clinical settings [22]. This review by Reychler et al. (2018) found that the use of IPV intervention in patients hospitalised with an acute exacerbation of COPD (n = 178) improved gas exchange ($PaO_2$ and $PaCO_2$) compared to various respiratory physiotherapy techniques and might reduce hospital LOS. Our systematic review is the first one to summarise the effectiveness of IPV intervention in the critical care population. The findings of our systematic review should be viewed with caution since there were various methodological (study design, outcome measures), clinical (patient population and application of IPV), and statistical (small sample size and lack of control group) heterogeneities observed. In addition, the interventions received by the comparator groups among the included studies also varied from "usual chest physiotherapy" [12] or "standard respiratory physiotherapy" [13] to "standard treatment," which only included medical management [15].

Studies that measured the effect of IPV on length of stay demonstrated some beneficial effects, where the median ICU-LOS appeared to be shortened by 1 to 2 days in the IPV group. However, significant heterogeneity was observed among the studies that reported on ICU-LOS. Two studies (Vargas et al. 2005 and Antonaglia et al. 2006) included patients with acute exacerbation of COPD whereas, Huynh et al. (2019) included upper abdominal and thoracic surgery patients. Huynh et al. (2019) did not find a significant reduction in ICU-LOS but reported a significant reduction in hospital LOS in the IPV group (IPV: 6.78 [4.98] vs. CPT: 8.40 [7.9] days p < 0.02). This outcome of hospital LOS, however, should be interpreted with caution as the non-randomised study design and the treatment frequency in the IPV group may introduce some bias. The duration and frequency of IPV intervention also varied among the included studies. A meta-analysis was performed, but due to the small number of studies and observed heterogeneity, it was not included in the main body of this review. Interestingly, the pooling of ICU-LOS data revealed that the magnitude and direction of the effect of IPV in reducing the ICU-LOS were similar in all three studies (S1 File).

Studies have reported that IPV improves gas exchange ($PaO_2$, $PaO_2/FiO_2$ ratio, and $PaCO_2$) in ventilated and non-ventilated patients [13, 14]. In this review, five out of six studies reported an improvement in gas exchange post IPV intervention [12–15, 27]. Notably, the time points of this outcome measurement varied among the studies; for example, Antonaglia et al. (2006) measured $PaO_2$ immediately prior to, and 30 minutes following the first IPV session and also at the time of discharge from ICU, whereas Tsuruta and colleagues (2006) recorded changes in $PaO_2$ at 3 hours interval for up to 24 hours [13, 14]. In contrast, Clini et al. (2006) measured the $PaO_2$ and $PaO_2/FiO_2$ ratio at five days intervals [12]. Despite lack of consistency across the studies, an improvement in oxygenation post IPV session(s) was found by the majority of the included studies. Short-term improvement in oxygenation could be attributed to the oxygen source which is used to drive the IPV device. Two studies reported the washout or stabilisation time prior to measuring oxygen levels [12, 13], whereas the time of measurement of oxygen levels in relation to IPV was not clear in other studies. Further, increases seen in oxygenation can be driven by the applied positive pressure, which may facilitate gas exchange by increasing overall functional residual capacity [28]. Also, positive pressure has been found to unload respiratory muscles, which subsequently reduces the oxygen cost of breathing, as demonstrated by Dimassi and colleagues (2011) [26], where the application of IPV led to a 20% reduction in diaphragm loading in post-extubation respiratory failure patients. In addition to these benefits, IPV can also augment oxygenation by promoting airway clearance. Some authors hypothesised that the improvement in oxygenation could, in part, be due to improved airway clearance post IPV intervention [20, 22]. In this review, three studies reported an observed increase in airway clearance with IPV intervention [12, 13, 15].

In addition to improved oxygenation, improved pulmonary ventilation has also been shown to reduce $PaCO_2$ levels in patients with COPD [29, 30] when treated with IPV. The applied positive pressure by IPV reduces the airway obstruction and thereby increases the expiratory flow in patients with airflow limitation secondary to COPD [29, 30]. A study in 25 patients with COPD demonstrated an increase in expiratory flow rate after IPV intervention (27). Studies also noted reduced $PaCO_2$ levels after application of IPV intervention in patients with COPD and patients with tracheostomy [12–15, 26]. Based on the available evidence, it appears that IPV may have a role in reducing $PaCO_2$ in hypercapnic respiratory failure patients.

The incidence of pneumonia in critically ill patients is well documented [3, 10, 31]. While there is some evidence that IPV may be effective as an airway clearance modality [18], its role in reducing or preventing the incidence of pneumonia in critically ill patients remains poorly studied. In this review, out of three studies [12, 13, 21], only one reported a significant

reduction in the incidence of pneumonia in the IPV group compared to the CPT group [12]. Surprisingly, the number of cases of pneumonia in all the included studies was very small in both the IPV and control groups. Although statistically significant, this small reduction in the incidence of pneumonia does not appear to be clinically meaningful. Evidence does not currently support the use of IPV to prevent or treat pneumonia in critically ill patients (S1 File). Further research is needed to evaluate the role of IPV in reducing the incidence of pneumonia. Similarly, despite airway clearance being one of the main indications of IPV intervention, only three studies reported an observed increase in secretion clearance. Surprisingly, none of these studies measured sputum yield [12, 13, 15]. Difficulties with accurate measurement of sputum yield have been well documented [32]. Due to the lack of data regarding airway clearance in the included studies, the role of IPV in airway clearance remains unclear.

Clinicians have been using IPV to improve lung volumes by recruiting partially or fully collapsed lung units [29, 30]. Despite this, the effect of IPV intervention in preventing or treating pulmonary atelectasis in acutely ill patients remains poorly researched. One reason may be that measuring the changes in pulmonary atelectasis can be challenging and expensive in a clinical setting. Deakins and Chatburn (2002) [33] used series of chest x-rays in paediatric patients, whereas Tsuruta and colleagues (2006) used chest x-rays and computed tomography scans [14]. Huynh et al. (2019) reported a significant reduction in several postoperative pulmonary complications, including atelectasis; however, it is unclear how this was assessed. In our review, only one study reported a resolution in compression atelectasis in mechanically ventilated patients [14]. Due to the small sample size of this observational study, the current level of evidence remains inconclusive regarding the role of IPV in treating pulmonary atelectasis in critically ill patients.

The clinical benefit of an intervention can be influenced by several factors such as adverse effects, poor treatment tolerance and patient compliance. The studies included in this review found IPV intervention to be safe. None of the studies reported any serious adverse events related to IPV intervention. A recent report of 35 critical care patients found that the application of IPV was safe in a critical care setting [34]. Furthermore, most of the studies in this review also reported that the IPV intervention was well tolerated by patients. However, the studies did not incorporate a subjective measure of tolerance of IPV intervention; instead, this was inferred from observation and treatment completion rates. Only one study, in 17 patients at risk of extubation failure, performed a subjective evaluation of IPV tolerance using a five-point scale ("severe discomfort" = 1 to "very good level of comfort" = 5). An average score of 3 ("acceptable level of comfort") was provided by 16 patients, whereas one patient found IPV to be very noisy [26]. Further studies are required that incorporate a subjective evaluation of the patient's experience with IPV intervention in critical care.

## Limitation

This systematic review has some limitations. The number of studies retrieved was small. While there is a chance that we were unable to find all the relevant studies, we minimised this by searching five databases and six trial registries for the last 40 years of publications. Heterogeneities resulting from differences in study design, patient population, dosage, and frequency of IPV intervention were frequently observed in the included studies. Further, small sample sizes and poor methodological quality introduces some bias and weakens the strength of conclusions of this review.

## Conclusions

This systematic review is the first to summarise the evidence of the IPV intervention in patients admitted to critical care. The findings of this review provide weak evidence to support

the use of IPV intervention in reducing ICU and hospital LOS, reducing respiratory rate, and improving gas exchange in critically ill patients. The therapeutic value of IPV in airway clearance and treating pulmonary atelectasis remains inconclusive, requiring further investigations. This review is based on a small number of available studies, mostly with small sample sizes. Hence, there is a need for more adequately powered randomised control trials to investigate the effectiveness of IPV intervention in improving outcomes such as ICU LOS, gas exchange, airway clearance, prevention or treatment of pneumonia and pulmonary atelectasis compared to routinely applied airway clearance and lung recruitment physiotherapy interventions in critical care population. In addition, there is also an indication for studies to evaluate patients' experiences with IPV intervention and their preference compared to routinely practiced respiratory physiotherapy interventions in critical care settings.

## Supporting information

**S1 Checklist.**
(DOCX)

**S1 Table. Search strategy.**
(DOCX)

**S1 File. Meta analysis.**
(DOCX)

**S2 File. Search results titles and abstracts.**
(XLSM)

**S3 File. Data extraction.**
(XLSX)

## Acknowledgments

We would like to thank Ms. Kanchana Ekanayake, Academic Liaison Librarian, The University of Sydney, for her assistance with the database search.

## Author Contributions

**Conceptualization:** Anwar Hassan, Jennifer Alison, Stephen Huang, Maree Milross.

**Data curation:** Anwar Hassan, William Lai, Maree Milross.

**Formal analysis:** Anwar Hassan, William Lai, Stephen Huang.

**Methodology:** Anwar Hassan, William Lai, Jennifer Alison, Stephen Huang, Maree Milross.

**Project administration:** Anwar Hassan.

**Software:** Anwar Hassan.

**Supervision:** Jennifer Alison, Stephen Huang, Maree Milross.

**Writing – original draft:** Anwar Hassan.

**Writing – review & editing:** Anwar Hassan, William Lai, Jennifer Alison, Stephen Huang, Maree Milross.

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
