## [Decision Letter · Decision Letter 0]

13 May 2021

PONE-D-21-11360

Effect of intrapulmonary percussive ventilation on intensive care unit length of stay, incidence of pneumonia and gas exchange in critically ill patients: a systematic review

PLOS ONE

Dear Dr. Hassan,

Thank you for submitting your manuscript to PLOS ONE. After careful consideration, we feel that it has merit but does not fully meet PLOS ONE’s publication criteria as it currently stands. Therefore, we invite you to submit a revised version of the manuscript that addresses the points raised during the review process.

In this instance it was extremely fortuitous that five content specific expert peer reviewers accepted to provide input to your manuscript. Consequently you will not to follow quite a deal of constructive feedback to assist with the evolving of the presenting of your work.

We look forward to receiving your revised manuscript.

Kind regards,

Shane Patman, PhD

Academic Editor

PLOS ONE

Journal Requirements:

2) Please include your tables as part of your main manuscript and remove the individual files. Please note that supplementary tables (should remain/ be uploaded) as separate "supporting information" files

3) Please include a copy of Table 3 which you refer to in your text. (We note you currently have two tables which are both titled 'Table 2'.)

4) Please include captions for your Supporting Information files at the end of your manuscript, and update any in-text citations to match accordingly. Please see our Supporting Information guidelines for more information: http://journals.plos.org/plosone/s/supporting-information.

5)  We note that you have indicated that data from this study are available upon request. PLOS only allows data to be available upon request if there are legal or ethical restrictions on sharing data publicly. For information on unacceptable data access restrictions, please see http://journals.plos.org/plosone/s/data-availability#loc-unacceptable-data-access-restrictions.

Reviewers' comments:

Reviewer's Responses to Questions

**Comments to the Author**

1. Is the manuscript technically sound, and do the data support the conclusions?

Reviewer #1: Yes

Reviewer #2: Yes

Reviewer #3: Partly

Reviewer #4: Yes

Reviewer #5: Partly

2. Has the statistical analysis been performed appropriately and rigorously? 

Reviewer #1: Yes

Reviewer #2: Yes

Reviewer #3: N/A

Reviewer #4: Yes

Reviewer #5: I Don't Know

3. Have the authors made all data underlying the findings in their manuscript fully available?

Reviewer #1: Yes

Reviewer #2: Yes

Reviewer #3: Yes

Reviewer #4: Yes

Reviewer #5: Yes

4. Is the manuscript presented in an intelligible fashion and written in standard English?

Reviewer #1: Yes

Reviewer #2: Yes

Reviewer #3: Yes

Reviewer #4: Yes

Reviewer #5: Yes

5. Review Comments to the Author

Reviewer #1: This manuscript reports a narrative review of reports on the effects of intrapulmonary percussive ventilation (IPV) applied to patients managed in intensive care units (ICU). Only 4 of the 7 included reports were randomised controlled trials; and only 2 studies achieved a PEDro score greater than 5; the overall risk of bias in these studies was moderate. The authors concluded “IPV may have a role in reducing ICU and hospital length of stay (LOS), reducing respiratory rate and improving gas exchange in critically ill adult patients.” (lines 539-541).

The methodology adopted for this systematic review is appropriate. However, it is my view that the clinical or therapeutic value of IPV was not sufficiently introduced and explained. The 7 studies included in this review compared IPV with ‘routine traditional chest physiotherapy’ or ‘standard medical care’, however I have two concerns with such comparisons and consider that the following points deserve discussion in the manuscript.

1- Routine traditional chest physiotherapy

Chest physiotherapy (CPT) is an old term which refers to a combination of secretion removal ‘techniques’. Furthermore, contemporary respiratory physiotherapy is no longer ‘routinely’ applied to patients. While the term CPT was the term used in the studies included in this review, the presentation of this current manuscript should reflect a clear understanding of the contemporary role of respiratory physiotherapy. Hence an analytical critique of the validity of using ‘CPT’ as a mechanical modality comparator should be included in the discussion.

2- Indications for and outcome measures of clinical intervention.

The effect of any intervention can only be appropriately reflected if the intervention employed follows relevant indications and is evaluated with appropriate outcome measures. The scientific value of a review will be improved if it includes discussion of the clinical implications of the reported ‘outcome measure(s)’. For example, is ‘reducing respiratory rate’ an expected positive outcome of IPV? If so, in what type of patients? The ‘Intervention’ subsection (lines 295-296) mentions that the “Most common indication to use IPV was to improve gas exchange, promote airway clearance and prevent or reverse pulmonary atelectasis”, but the only focused outcome measure reported in the studies reviewed was gas exchange. It should be noted that IPV is in fact a high frequency percussive ventilation mode and the ventilator is driven by an oxygen source, so it is not surprising that this intervention could result in a short duration of increased oxygenation. How then should a marginal improvement in oxygenation for a short duration after an intervention be interpreted? What are the clinical implications and clinical value of gas exchange improvement for a brief period after application of this mode of ventilation? I believe readers of this manuscript would expect a more thorough discussion of the clinical indications for ‘therapeutic’ IPV and how these indications could be met.

I would like to suggest:

1. The ‘Introduction’ section be strengthened by including a more detailed explanation of the proposed ‘therapeutic’ mechanism of IPV, and what IPV is supposed to achieve. For example, how might the ‘Birdian flow’ generated by IPV, facilitate lung expansion and secretion removal. Lines 139-144 state that “IPV is used in a range of patients…”. Please include the reasons and who recommended such use, and what were the reported ‘outcomes’ in patients with COPD, burns and thoracic surgery.

2. Include the discussion of the 2 concerns raised above.

3. The conclusion be reworded to: ‘the evidence to support IPV’s role in reducing ICU and hospital LOS, and improvement in gas exchange is weak. The therapeutic value of IPV in secretion removal and lung recruitment requires further investigation’.

Reviewer #2: Overall this is a well done systematic review.

There are few minor points to query

At 461 the reason for the improvement in oxygenation and PaCO2 is discussed. The reasons suggested are also facilitated by standard mechanical ventilation eg increase in FRC, unloading of diaphragm and also modalities such as NIV. Why would the modality of either IPV added to MV or used instead of NIV in a non ventilated patient have benefits?. There have been more specific or physiological reasons suggested which I think should be discussed in more depth.

486-487 – Grammar could be improved in this sentence

Reviewer #3: The authors have conducted a systematic review process on an interesting topic of IPV use in the critical care environment. While most of the methodology appears sound, there are some areas for clarification / improvement. Analysis of reported results appears to have mistakes or be incorrectly interpreted. Overall, this has led to the authors painting a positive slant on the findings, whereas the results appear more disparate and combined with the high risk of bias…..probably doesn’t reflect being able to report that it may reduce LOS, improve gas exchange and reduce RR.

Line 86: Out of 306 studies. Consider deleting or clarifying that 306 was number resultant from the search e.g. Out of 306 identified abstracts,

Line 81, 169. If an update of the search has been done to include up to 2021, this should be reflected here (rather than 2019)

Line 141. Consider deleting “few”.

Line 148, 151, 174-175. Consider better and consistent terms. :non-critically ill stable patients”. Maybe most studies were stable patients on ward rather than critical care settings. Avoid interchanging between critically ill and acutely ill.

Line 181. Therapeutic purposes. Continuous mechanical ventilation would be a therapeutic purpose. Consider another term to define the applications you were after e.g. its intermittent application for goals of airway clearance or lung recruitment.

Line 184-188. Some of these, but probably not all would have been of primary interest i.e. set primary & secondary outcomes; but some arisen from what was reported in the papers. Please clarify. You would have had a primary outcome measure/aim to determine, and also have the methods indicate you recorded outcome measures from across all studies. In your results then you will report that you found that not enough papers reported this to be able to do a metaanalysis and you report the other outcome measures found (as per section 312-318). Should align to statements in 260-265 e.g. where pneumonia is mentioned, but not atelectasis which the earlier section reported as an inclusion. Please clarify / revise.

Line 214, 215. Cochrane and PEDro scales presented, then statement of moderate risk of bias. “Moderate” is not a listed category in either of the scales as presented.

Line 281-282, 288. Are these needed? It was essentially the inclusion criteria. Reporting the % of patients on each device might be more of interest.

Line 295-296. Were these indications stated in the papers or is this the authors’ description of the indications to use the equipment? Please clarify / confirm.

Greater use of referencing and or referrals to Tables is needed throughout the results section so it is clear where results came from e.g. lines 308-309

Line 341, 347. Revise sentences

Line 348. Data presented show an higher incidence of pneumonia in the IPV group, but the discussion is indicating the IPV group was lower. Please check / revise.

The results and conclusions seem biased towards indicating some potential benefit. Data has not been able to be pooled though. The associations of possible affect seem over-emphasised.

LOS – Largest study result NSD, but reported as shorter.

Pneumonia – only 1 of 3 studies showed a significant change

PaO2. Lines 363-364; 380-381. These do not appear to be between group comparisons reporting significance. Both groups improved oxygenation across the period.

Discussion. The low quality of the studies included should be emphasised at the onset. Start with lines 425-431. Discussion will need significant review once results of study reviewed. Avoid repeating results in the discussion.

Reviewer #4: This systematic review investigates the Effect of intrapulmonary percussive ventilation on intensive care unit length of stay, incidence of pneumonia and gas exchange in critically ill patients. My only criticism relates to the broad inclusion criteria " Studies that reported the effects of IPV, high-frequency ventilation, and high-frequency oscillation where these interventions were primarily used for therapeutic purposes were included,...". IPV is not the same as high frequency ventilation or high frequency oscillation. HFV and HFO may often be applied via oscillatory vests which are not the same as IPV delivered via a face mask/mouthpiece or artificial airway.

Reviewer #5: Effect of intrapulmonary percussive ventilation on intensive care unit length of stay,

incidence of pneumonia and gas exchange in critically ill patients: a systematic review.

Thank you for the opportunity to review this manuscript. The topic is interesting and relevant for clinical practice. I have some comments that the authors may wish to consider when making revisions. These link to the abstract and main paper.

Introduction

The title implies an interest in outcomes relating to presence or absence of pneumonia and components of gas exchange as well as LOS.

Aims and research question have more consistent outcomes outlined; and lines 152-154 more detail.

Could the explicit aims of IPV be provided so that these could considered with this review of literature and resulting aims; this would also then link to the discussion again later (somewhat addressed in lines 131-136)

Information about IPV use in the literature is presented. The study by Reychler 2018 evaluated airway clearance and gas exchange; was the current review not interested in airway clearance too (albeit in critical care settings)?

150/151 Care- the role of IPV in preventing or reversing atelectasis is possibly only one only unanswered question.

Line 134: The term conventional chest physiotherapy (ref 12) is used; is there an explanation of this please.

Line 263- is atelectasis one of the outcomes of interest? it seems so

Lines 264, 265 – this seems in conflict to results presented in the supplementary files

Methods

Patients > or =16years – is this a consistent international cut off for adult critical care units? I am curious that there is potential for studies (included or excluded) conducted in a mixed peadiatric:adult population dependent on the age cut off.

The mixed study design does detract somewhat from the review and it may be useful to group the presentation of results from the RCTs/quasi randomised and discuss the weight of evidence relating to these.

Additionally, is it possible to appraise and accurately score the quality of an abstract or a non RCT using the PEDro scale which is designed for the quality appraisal of a fully reported RCT(Lines 213 a-215). There are other tools available for appraisal of non RCTs.

Same with Cochrane ROB tool – how can an observational study score under the item “random sequence generation”.

Results

It may be worth considering the weight/magnitude of the results relating to outcomes from the RCTs; and then presenting the results from the observational studies separately. Otherwise, there is a nice overview of outcome measures included overall (Lines 311-318 onwards), followed by relevant results sections.

294 presents an overview of the intervention (dose, interface etc). there should be a summary of the control group/comparator groups used.

295/296 – interesting that use of IPV relating to airway clearance is included here, but limited reference to this earlier.

Lines 361-361- unsure of the term “most significant” here

Table 1 & 2 – as per methods please consider the suitability of the Cochrane ROB and the PEDro scale for non RCTs.

Discussion

Were there any papers which overlapped with the Reychler 2018 study?

Line 425 what does routine chest physiotherapy refer to here; please consider the term routine, or use quotations if this is directly from the publication.

430/431 refer to previous comment in terms of judging ROB and quality.

456-459 This is confusing as the review indicated that meta analysis not possible due to heterogeneity. In the supplementary material the I squared statistic does indeed support this high heterogeneity; this implies uncertainty about this specific result and its interpretation (i.e. from the supplementary material).

481-407 Could this discussion be more concise. a/a further pooled data yet this is not reported in the results as according to the methods meta analysis was not feasible.

Line 486 Check the word “don’t”!

The discussion is quite long.

Perhaps consider a paragraph summarising the clinical implications in terms of patient selection, intervention (timing and dose) etc or add this to the conclusion. The review sates “more clinical trials with larger sample sizes are warranted to further add to the findings of this review”; this is a very broad statement. Could the authors consider a more focused suggestion to guide the next steps for future research e.g. using the PICO to frame this; It would be interesting to debate what study design should be considered

6. PLOS authors have the option to publish the peer review history of their article (what does this mean?). If published, this will include your full peer review and any attached files.

Reviewer #1: **Yes: **Alice YM Jones

Reviewer #2: No

Reviewer #3: No

Reviewer #4: **Yes: **George Ntoumenopoulos

Reviewer #5: No

---

## [Author Response · Author response to Decision Letter 0]

26 Jun 2021

Response to the editor

Dear Editor,

Thank you for the opportunity to revise our manuscript. Our responses to the editors’ and reviewers’ comments are outlined below under each comment. Since there were numerous comments, for clarity, we have left the comments in black font (as received), and used blue font for our responses. In addition, we have used green font in italics with quotation marks to show the sections that has been amended in the manuscript. All the responses have page number, the heading and subheading and the line numbers, so that it can be easily located in the revised manuscript. The line numbers indicated in our responses match with the line numbers in the unmarked version (“Manuscript”) and not the tracked version. So please refer to the clean version when going through the following responses. Tracked version is also provided as per your request. Thank you.

Our responses are as follows;

Response: The manuscript (Title page and main body) is prepared according to PLOS ONE style. 

2) Please include your tables as part of your main manuscript and remove the individual files. Please note that supplementary tables (should remain/ be uploaded) as separate "supporting information" files

Response: All the tables (Table 1, Table 2 and Table 3) are included/embedded in the main body of the manuscript.

3) Please include a copy of Table 3 which you refer to in your text. (We note you currently have two tables which are both titled 'Table 2'.)

Response: Table 3 is included now in the body of manuscript as per your request. 

4) Please include captions for your Supporting Information files at the end of your manuscript, and update any in-text citations to match accordingly. Please see our Supporting Information guidelines for more information.

Response: Captions have been provided at the end of the manuscript as per the journal guidelines. 

5) We note that you have indicated that data from this study are available upon request. PLOS only allows data to be available upon request if there are legal or ethical restrictions on sharing data publicly

Response: During the first submission, we stated that information will be provided upon request (we meant journal’s request). Apologies if this caused any confusion. Since this is a systematic review, we do not have patient related data. There are no restrictions with providing any information related to this systematic review. The search strategy (S1 Table) has been uploaded already which will allow replicating the search. In addition, please let us know if you require any additional information. 

 Response to reviewer’s comments

Review Comments to the Author

Reviewer #1: This manuscript reports a narrative review of reports on the effects of intrapulmonary percussive ventilation (IPV) applied to patients managed in intensive care units (ICU). Only 4 of the 7 included reports were randomised controlled trials; and only 2 studies achieved a PEDro score greater than 5; the overall risk of bias in these studies was moderate. The authors concluded “IPV may have a role in reducing ICU and hospital length of stay (LOS), reducing respiratory rate and improving gas exchange in critically ill adult patients.” (lines 539-541).

The methodology adopted for this systematic review is appropriate. However, it is my view that the clinical or therapeutic value of IPV was not sufficiently introduced and explained. The 7 studies included in this review compared IPV with ‘routine traditional chest physiotherapy’ or ‘standard medical care’, however I have two concerns with such comparisons and consider that the following points deserve discussion in the manuscript.

1- Routine traditional chest physiotherapy

Chest physiotherapy (CPT) is an old term which refers to a combination of secretion removal ‘techniques’. Furthermore, contemporary respiratory physiotherapy is no longer ‘routinely’ applied to patients. While the term CPT was the term used in the studies included in this review, the presentation of this current manuscript should reflect a clear understanding of the contemporary role of respiratory physiotherapy. Hence an analytical critique of the validity of using ‘CPT’ as a mechanical modality comparator should be included in the discussion.

Response: Thank you for the helpful suggestion regarding clarifying terminology used. In the included studies, terms such as “chest physiotherapy” (CPT) and “standard respiratory physiotherapy” have been used by the authors. For consistency throughout the manuscript, we have used the term “chest physiotherapy” has been used in this systematic review. However, there was one instance where the term “conventional chest physiotherapy” was used which has now been replaced by “chest physiotherapy”. Also, based on reviewer’s suggestion, further explanation and clarification of “chest physiotherapy” has been added in the introduction and result section. 

Please see page 5, Introduction: Lines128-132 “Chest physiotherapy interventions (CPT) such as chest percussion & vibrations, postural drainage, positioning, thoracic expansion exercises, manual hyperinflation, ventilator hyperinflation and airway suctioning aim to promote airway secretion clearance, increase alveolar recruitment…” 

Please see page 18, Results/Intervention: Lines 296-298 “CPT included chest clapping, postural drainage, expiration with open glottis, incentive spirometer and mobilisation (Table 3). Duration of CPT session ranged from 30 minutes to 60 minutes once or twice a day”

2- Indications for and outcome measures of clinical intervention.

The effect of any intervention can only be appropriately reflected if the intervention employed follows relevant indications and is evaluated with appropriate outcome measures. The scientific value of a review will be improved if it includes discussion of the clinical implications of the reported ‘outcome measure(s)’. For example, is ‘reducing respiratory rate’ an expected positive outcome of IPV? If so, in what type of patients? The ‘Intervention’ subsection (lines 295-296) mentions that the “Most common indication to use IPV was to improve gas exchange, promote airway clearance and prevent or reverse pulmonary atelectasis”, but the only focused outcome measure reported in the studies reviewed was gas exchange. It should be noted that IPV is in fact a high frequency percussive ventilation mode and the ventilator is driven by an oxygen source, so it is not surprising that this intervention could result in a short duration of increased oxygenation. How then should a marginal improvement in oxygenation for a short duration after an intervention be interpreted? What are the clinical implications and clinical value of gas exchange improvement for a brief period after application of this mode of ventilation? I believe readers of this manuscript would expect a more thorough discussion of the clinical indications for ‘therapeutic’ IPV and how these indications could be met.

Response: Thank you for reviewer’s valuable input and suggestions. There are multiple questions in the above comment, so we have responded to them in separate sections.

Respiratory rate can be high in critical care patients with excessive secretions or underlying pulmonary atelectasis. IPV is mainly used to remove bronchial secretions and recruit lung units thereby improving gas exchange with the potential corollary of a reduction in respiratory rate. Although IPV is primarily not used for treating tachypnoea, reduction in respiratory rate is a surrogate measure of improved respiratory status hence it is important to measure this. The changes in respiratory rate in the included studies are very small and we agree that they are not clinically meaningful. This has been acknowledged in the manuscript. 

Please see page 22, Results/Respiratory rate: Lines 386-388“Overall, three out of four studies reported a small but significant reduction in respiratory rate post IPV intervention. This small change does not seem to be clinically relevant.”

Further, we agree with the reviewer’s observation that although IPV may have a role in secretion clearance, gas exchange and pulmonary atelectasis, the commonly reported and measured outcome in the included studies was “gas exchange”. Three studies reported an observed improvement in secretion clearance, but this was not measured. We have acknowledged this. 

Please see page 22, Results/Airway clearance: Lines 391-393 “Three studies in this review reported an observed increase in airway clearance with IPV intervention; however, none of them measured the expectorated sputum weight (wet or dry)(12,13,15) or other measures of mucociliary clearance”.

In addition, included studies have also reported on atelectasis, respiratory rate, and pneumonia as outcomes which are reported in results and in the Discussion section. We have also reported and discussed airway clearance in the results and discussion section. 

Please see page 22, Results/Airway clearance: lines 391-393 as above “Three studies in this review reported an observed increase in airway clearance with IPV intervention; however, none of them measured the expectorated sputum weight (wet or dry)(12,13,15) or other measures of mucous clearance”.

Page 24, Discussion: lines 458-460 “Some authors hypothesised that the improvement in oxygenation could, in part, be due to improved airway clearance post IPV intervention(20,22). In this review, three studies reported an observed increase in airway clearance with IPV intervention”

It is true that oxygen is the driving gas for IPV device, which may cause a transient increase in the oxygen levels. A washout or stabilisation period of 15-30 minutes is recommended before measurement of oxygen. In the included studies, some studies (Antonaglia et al. 2006 and Clini et al. 2006) have reported a stabilisation period whereas this was not clear in other studies which may introduce some bias. This has now been mentioned and discussed in the ‘discussion” section. 

Pease see page 24, Discussion: Lines 448-451 “Short term improvement in oxygenation could be attributed to the oxygen source which is used to drive the IPV device. Two studies reported the washout or stabilisation time prior to measuring oxygen levels(12,13) whereas the time of measurement of oxygen levels in relation to IPV was not clear in other studies”

I would like to suggest:

1. The ‘Introduction’ section be strengthened by including a more detailed explanation of the proposed ‘therapeutic’ mechanism of IPV, and what IPV is supposed to achieve. For example, how might the ‘Birdian flow’ generated by IPV, facilitate lung expansion and secretion removal. 

Response: The reviewer’s suggestion has been taken into consideration and further details have been added in the “Introduction” section. 

Please see page 5, Introduction: Lines 136-143 “IPV is a non-continuous form of high-frequency ventilation delivered by a pneumatic device that provides small bursts of sub-physiological tidal breaths at a frequency of 60 - 600 cycles/minute superimposed on a patient's breathing cycle(16–18). The high frequency breaths create shear forces causing dislodgement of the airway secretions. Furthermore, the IPV breath cycle has an asymmetrical flow pattern characterised by larger expiratory flow rates, which may propel the airway secretions towards the central airway(18). In addition, the applied positive pressure recruits the lung units, resulting in a more homogeneous distribution of ventilation and improved gas exchange”

Lines 139-144 state that “IPV is used in a range of patients…”. Please include the reasons and who recommended such use, and what were the reported ‘outcomes’ in patients with COPD, burns and thoracic surgery.

Response: We have now added the requested details about the types of clinical conditions and indications (with citations). 

Please see page 5, Introduction: Lines 143-148 “In acute care and critical care settings, IPV intervention is used in a range of patients, from spontaneously breathing patients to those receiving invasive mechanical ventilation where IPV breaths can be superimposed on a patient’s breathing cycle or superimposed on breaths delivered by a mechanical ventilator. The most common indications for IPV use are reported as removal of excessive bronchial secretions, improving gas exchange, and recruitment of atelectatic lung segments(12–14,18).”

2. Include the discussion of the 2 concerns raised above.

Response: Concerns addressed. 

Please see page 23, Discussion: Lines 419-422 “In addition the interventions received by the control groups among the included studies also varied from “usual chest physiotherapy”(12) or “standard respiratory physiotherapy”(13) to “standard treatment” which included medical management(15)”. 

Page 24, Discussion: Lines 458-459 “Some authors hypothesised that the improvement in oxygenation could, in part, be due to improved airway clearance post IPV intervention(20,22).”

Page 24, Discussion: Lines 448-451 “Short term improvement in oxygenation could be attributed to the oxygen source which is used to drive the IPV device. Two studies have reported the washout or stabilisation time prior to measuring oxygen levels(12,13) whereas it was not clear in other studies.”

3. The conclusion be reworded to: ‘the evidence to support IPV’s role in reducing ICU and hospital LOS, and improvement in gas exchange is weak. The therapeutic value of IPV in secretion removal and lung recruitment requires further investigation’.

Response: The “Conclusions” section has been revised and reworded as per reviewer’s suggestions. 

Please see page 27, Conclusions: lines 524-528 “The findings of this review provide weak evidence to support the use of IPV intervention in reducing ICU and hospital LOS, reducing respiratory rate, and improving gas exchange in critically ill patients. The therapeutic value of IPV in airway clearance and treating pulmonary atelectasis remains inconclusive, requiring further investigations.”

Reviewer #2: Overall this is a well done systematic review.

There are few minor points to query

At 461 the reason for the improvement in oxygenation and PaCO2 is discussed. The reasons suggested are also facilitated by standard mechanical ventilation eg increase in FRC, unloading of diaphragm and also modalities such as NIV. Why would the modality of either IPV added to MV or used instead of NIV in a non ventilated patient have benefits?. There have been more specific or physiological reasons suggested which I think should be discussed in more depth.

Response: Participants in the cited studies who receive invasive and non-invasive mechanical ventilation will have similar physiological benefits such as increase in FRC and partial unloading of the diaphragm, However, IPV has airway clearance properties (explained in the Introduction section) which may have an added advantage over mechanical ventilation. This is now discussed in the Introduction and Discussion section as below; 

Please see page 5, Introduction: Lines 136-141 “IPV is a non-continuous form of high-frequency ventilation delivered by a pneumatic device that provides small bursts of sub-physiological tidal breaths at a frequency of 60 - 600 cycles/minute superimposed on a patient's breathing cycle(16–18). The high frequency breaths create shear forces causing dislodgement of the airway secretions. Furthermore, the IPV breath cycle has an asymmetrical flow pattern characterised by larger expiratory flow rates, which may propel the airway secretions towards the central airway”

Please see page 24, Discussion: Lines 453-459 “Also, positive pressure has been found to unload respiratory muscles which subsequently reduces the oxygen cost of breathing, as demonstrated by Dimassi and colleagues (2011)(26) where the application of IPV led to a 20% reduction in diaphragm loading in post-extubation respiratory failure patients. In addition to these benefits, IPV can also augment oxygenation by promoting airway clearance. Some authors hypothesised that the improvement in oxygenation could, in part, be due to improved airway clearance post IPV intervention.”

486-487 – Grammar could be improved in this sentence

Response: We have checked the grammar and also the word “don’t” is now replaced by “does not”. 

Please see page 25, Discussion: Line 475-476. “Although statistically significant, this small reduction in pneumonia does not appear to be clinically meaningful”.

Reviewer #3: The authors have conducted a systematic review process on an interesting topic of IPV use in the critical care environment. While most of the methodology appears sound, there are some areas for clarification / improvement. Analysis of reported results appears to have mistakes or be incorrectly interpreted. Overall, this has led to the authors painting a positive slant on the findings, whereas the results appear more disparate and combined with the high risk of bias…..probably doesn’t reflect being able to report that it may reduce LOS, improve gas exchange and reduce RR.

Response: Thank you for the reviewer’s detailed interpretation and valuable suggestions on this manuscript. We have reviewed all the analyses and made some amendments in the interpretation of the findings of this review that are now mentioned in both the Discussion and Conclusions section. Please see our responses below to specific points made by this reviewer;

Please see page 22, Discussion: Lines 406-409 “The findings of this review provide weak evidence to support the effectiveness of IPV intervention in reducing ICU-LOS, improving gas exchange and reducing respiratory rate in critically ill patients compared to chest physiotherapy techniques or standard medical management”.

Please see page 27, Conclusions: Lines 523-528 “This systematic review is the first to summarise the evidence of the IPV intervention in patients admitted to critical care. The findings of this review provide weak evidence to support the use of IPV intervention in reducing ICU and hospital LOS, reducing respiratory rate, and improving gas exchange in critically ill patients. The therapeutic value of IPV in airway clearance and treating pulmonary atelectasis remains inconclusive, requiring further investigations.”

Line 86: Out of 306 studies. Consider deleting or clarifying that 306 was number resultant from the search e.g. Out of 306 identified abstracts,

Response: This has been amended as per reviewer’s suggestion. 

Please see page 3, Abstract/Results: Line 85 “Out of 306 identified abstracts, seven studies (630 patients) met the eligibility criteria”.

Line 81, 169. If an update of the search has been done to include up to 2021, this should be reflected here (rather than 2019)

Response: This has now been updated to “2021” in both the Abstract and in Methods/Search Strategy section. 

Please see page 3, Abstract/Methods: Lines 77- 78 “A systematic search of intrapulmonary percussive ventilation in intensive care unit (ICU) was performed on five databases from 1979 to 2021” 

Page 7, Methods/search strategy: Lines 171-173 “The first stage included database searches on MEDLINE, EMBASE, CINAHL, Web of Science, and PEDro, from 1979 (when IPV was first introduced) to February 2021”

Line 141. Consider deleting “few”.

Response: Deleted “few” as per the reviewer’s suggestion. It read “In the last two decade, few studies have reported….”, now after amendment, it reads “In the last two decades, studies have reported IPV in the critical….” Please see page 6, Introduction: line 149

Line 148, 151, 174-175. Consider better and consistent terms. :non-critically ill stable patients”. Maybe most studies were stable patients on ward rather than critical care settings. Avoid interchanging between critically ill and acutely ill.

Response: In response to the reviewer’s comment, the terms “non-critically ill” and “non-acutely ill” has been replaced with “stable patients” and “critically ill patients” throughout the manuscript. 

Please see page 6, Introduction, Lines 157-158 “Most of the studies reviewed by Reychler and colleagues (2018) included stable patients”

Page 6 Introduction: Lines 155-157. “The question regarding the role of IPV in preventing or reversing atelectasis and reducing the incidence of pneumonia in critically ill patients remains unanswered”

Page 7, Methods/Inclusion and exclusion criteria: Lines 183-184 “Studies that included stable patients in the inpatient, outpatient, or community-based settings were excluded.” 

Line 181. Therapeutic purposes. Continuous mechanical ventilation would be a therapeutic purpose. Consider another term to define the applications you were after e.g. its intermittent application for goals of airway clearance or lung recruitment.

Response: This has been amended as per reviewer’s suggestion,

Please see page 7, Methods/Inclusion criteria: Lines 187-192 “Studies that reported the effects of IPV, high-frequency ventilation, and high-frequency oscillation where these interventions were primarily used intermittently for short duration to promote airway clearance, reverse or treat pulmonary atelectasis, or to improve gas exchange were included, whereas the studies that used these interventions to provide continuous mechanical ventilation were excluded.”

Line 184-188. Some of these, but probably not all would have been of primary interest i.e. set primary & secondary outcomes; but some arisen from what was reported in the papers. Please clarify. You would have had a primary outcome measure/aim to determine, and also have the methods indicate you recorded outcome measures from across all studies. In your results then you will report that you found that not enough papers reported this to be able to do a metaanalysis and you report the other outcome measures found (as per section 312-318). Should align to statements in 260-265 e.g. where pneumonia is mentioned, but not atelectasis which the earlier section reported as an inclusion. Please clarify / revise.

Response: As per reviewer #3 suggestion, we have clarified our primary and secondary outcomes of interest, “atelectasis” is now added. 

Please see page 12, Methods/outcome measures: Lines 243- 246 “The primary outcome of interest was ICU-LOS. Secondary outcomes included PaO2, the ratio of the partial pressure of arterial oxygen and fraction of inspired oxygen (PaO2/FiO2), PaCO2, airway clearance, the incidence of pneumonia, respiratory rate, and pulmonary atelectasis.”

Line 214, 215. Cochrane and PEDro scales presented, then statement of moderate risk of bias. “Moderate” is not a listed category in either of the scales as presented.

Response: In response to the reviewer’s comment, we have amended our description of the risk of bias. 

Please see page 9, Methods: Lines 224-227 “Overall, based on the Cochrane assessment tool, three out of four studies appear to have a low risk of bias whereas in one study the risk of bias was high. On the PEDro scale, the quality of the studies ranged from “poor” to “good.” 

Line 281-282, 288. Are these needed? It was essentially the inclusion criteria. Reporting the % of patients on each device might be more of interest.

Response: As per reviewer’s suggestion we have deleted the following sentence 

“All the studies were conducted in the critical care setting where patients were receiving invasive mechanical ventilation, non-invasive ventilation, or were breathing spontaneously with supplemental oxygen.” 

And have added “Study sample sizes ranged from 10 patients to 419 patients(14,21) (Table 3). All the studies were conducted in the critical care setting which included patients who were mechanically ventilated (34%), post extubation (6%), requiring NIV (18%) and the remaining (42%) were requiring high oxygen therapy (FiO2 ≥ 40%) and continuous positive airway pressure support.” (See page 17, Results/Study characteristics: Lines 266-270)

Line 295-296. Were these indications stated in the papers or is this the authors’ description of the indications to use the equipment? Please clarify / confirm.

Response: These indications were used by the authors of the included studies. This has now been made clearer. 

Please see page 17, Results/Intervention: line 280-282 “In the included studies, most common indications to use IPV intervention were to improve gas exchange, promote airway clearance and prevent or reverse pulmonary atelectasis.”

Greater use of referencing and or referrals to Tables is needed throughout the results section so it is clear where results came from e.g. lines 308-309

Response: Point noted with thanks. Tables have now been referenced more frequently throughout the Results section as per the reviewer’s suggestion. Please see page 18, Methods/Intervention: Lines 286-299.

Line 341, 347. Revise sentences

Response: The sentences have been revised as requested. 

(i) Please see page 19, Results/Length of stay: Lines 329-332 “Another study by Vargas et al. (2005) investigated the effects of IPV intervention in 33 patients with COPD with acute respiratory failure where the intervention group (n = 17) received IPV for 30 minutes twice a day, and the control group (n =16) received standard medical treatment.”

(ii) Please see page 20, Results/Incidence of pneumonia: Lines 338-340 “Antonaglia et al. (2006) reported a small (not significant) difference in the incidence of pneumonia in 40 patients with acute exacerbation of COPD (IPV: 2 vs. CPT: 4, NS).”

Line 348. Data presented show an higher incidence of pneumonia in the IPV group, but the discussion is indicating the IPV group was lower. Please check / revise.

Response: Noted with thanks. This was a typographical error which has been corrected.

 Please see page 20, Results/Incidence of pneumonia: Lines 338-340 “Antonaglia et al. (2006) reported a small (not significant) difference in the incidence of pneumonia in 40 patients with acute exacerbation of COPD (IPV: 2 vs. CPT: 4, NS).”

The results and conclusions seem biased towards indicating some potential benefit. Data has not been able to be pooled though. The associations of possible affect seem over-emphasised.

LOS – Largest study result NSD, but reported as shorter. 

Response: Thanks for reviewer’s valuable feedback, it has been mentioned in the manuscript that the reduction in the LOS in Huynh et al (2019) study is shorter but statistically not significant. (Please see page19, Results/Length of stay, Lines 325:327). 

We also acknowledge reviewer’s point about data pooling. For the LOS data, we performed a meta-analysis (supplied as supplementary, see “S1 File”) where the average median difference of all the three studies showed a shorter LOS with a p-value <0.01(Antonaglia et al. 2006, Vargas et al. 2006 and Huynh et al. 2019). However due to small number of studies the meta-analysis is not included in the main body of the manuscript. This may have influenced our conclusions; however, we agree that the risk of bias remains high in most studies which adds another perspective to the outcomes. Based on the reviewer’s feedback we have made some amendments to the study conclusions. 

Please see page 27, Conclusions: Lines 523-528. “This systematic review is the first to summarise the evidence of the IPV intervention in patients admitted to critical care. The findings of this review provide weak evidence to support the use of IPV intervention in reducing ICU and hospital LOS, reducing respiratory rate, and improving gas exchange in critically ill patients. The therapeutic value of IPV in airway clearance and treating pulmonary atelectasis remains inconclusive, requiring further investigations.”

Pneumonia – only 1 of 3 studies showed a significant change.

Response: As the reviewer correctly notes, only 1 in 3 studies showed difference in the incidence of pneumonia with the use of IPV. This is reflected in our conclusion that IPV does not seem to be effective in preventing or reducing the incidence of pneumonia. 

Please see page 25, Discussion: Lines 472-478“In this review, out of three studies(12,13,21), only one reported a significant reduction in the incidence of pneumonia in the IPV group compared to the CPT group(12). Although statistically significant, this small reduction in pneumonia does not appear to be clinically meaningful. Surprisingly, the number of cases of pneumonia in all the included studies was very small in both the IPV and control groups. Evidence does not currently support the use of IPV to prevent or treat pneumonia in critically ill patients”

PaO2. Lines 363-364; 380-381. These do not appear to be between group comparisons reporting significance. Both groups improved oxygenation across the period.

Response: Thank you for pointing this out and we apologise for causing confusion here by inserting the p-value twice (typographical error) which doesn’t reflect the actual study findings by Antonaglia et al 2006. The comparison is in fact `between group` comparisons (Please see Antonaglia et al 2006, Table 4, page 2942). This error has been rectified by removing one of the p-values from PaO2/ FiO2 and PaCO2 results. 

Please see page 20, Results/ PaO2 and PaO2/FiO2 ratio: Lines 350-352 “Antonaglia et al. (2006) reported a significant change in PaO2/ FiO2 ratio from admission to discharge (seven days) in patients with COPD admitted to ICU (IPV: 173 �27] to 274 �15], Control: 181 �29] to 237 �20], p < 0.01)”.

Page 21, Results/Change in PaCO2: Lines 366-368 “Antonaglia et al. (2006) recorded a significant reduction in PaCO2 levels in patients with COPD in IPV and CPT group (IPV: 79 �7] to 58 �5.4], Control: 80��6.5] to 64 �5.2] mmHg, p < 0.01)”

Discussion. The low quality of the studies included should be emphasised at the onset. Start with lines 425-431. Discussion will need significant review once results of study reviewed. Avoid repeating results in the discussion.

Response: As per reviewer’s suggestion, we have now highlighted the quality of papers and risk of bias in the methods, results and discussion section. 

(1) Please see page 12, Methods/Outcome measures: Lines 247-248 “Due to the small number of studies and observed heterogeneities in the study methodology and patient population, all the outcomes were summarised narratively”

(2) Please see page 23, Discussion: Lines 416-419 “The findings of our systematic review should be viewed with caution since there were various methodological (study design, outcome measures), clinical (patient population and application of IPV) and statistical (small sample size and lack of control group) heterogeneities observed.”

(3) Please see page 27, Limitations: Lines 517-520 “Heterogeneities resulting from differences in study design, patient population, dosage, and frequency of IPV intervention were frequently observed in the included studies. Further, small sample sizes and poor methodological quality introduces some bias and weakens the strength of conclusions of this review.”

Reviewer #4: This systematic review investigates the Effect of intrapulmonary percussive ventilation on intensive care unit length of stay, incidence of pneumonia and gas exchange in critically ill patients. My only criticism relates to the broad inclusion criteria " Studies that reported the effects of IPV, high-frequency ventilation, and high-frequency oscillation where these interventions were primarily used for therapeutic purposes were included,...". IPV is not the same as high frequency ventilation or high frequency oscillation. HFV and HFO may often be applied via oscillatory vests which are not the same as IPV delivered via a face mask/mouthpiece or artificial airway.

Response: As reviewer 4 notes, our broad inclusion criteria may have initially identified irrelevant studies. 

IPV has been used clinically for more than four decades and yet it remains poorly defined. Clinical classification (based on the breath frequency) of different high frequency ventilators (including IPV) is unclear (Kallet et al 2013: Resp Care, Vargas et al 2005: Crit Care, Salim et al 2005: Crit Care Med). In most of the studies, IPV remains the most used terminology, however as reviewer 4 notes, infrequently, terminologies such as HFV (High frequency ventilation) and HFO (High frequency oscillation) and CHFO (Chest high frequency oscillation) have also been used interchangeably. Hence, to ensure we retrieved all the relevant studies in our review, we included these terminologies in our search (please see the supplementary table: Table 1S: Search strategy). These retrieved studies were then screened carefully according to our selection criteria.

As correctly noted by the reviewer that HFV can be applied via a vest (worn by patients) also commonly known as High Frequency Chest Wall Oscillation (HFCWO). HFCWO is applied externally via a vest whereas IPV is applied internally via airways achieving different physiological effects via completely different mechanisms. Hence, HFCWO was not used in our search strategy. All studies identified by the search strategy were then screened according to our selection criteria to ensure only studies of IPV were included in our systematic review. 

Reviewer #5: Effect of intrapulmonary percussive ventilation on intensive care unit length of stay, incidence of pneumonia and gas exchange in critically ill patients: a systematic review.

Thank you for the opportunity to review this manuscript. The topic is interesting and relevant for clinical practice. I have some comments that the authors may wish to consider when making revisions. These link to the abstract and main paper.

Introduction

The title implies an interest in outcomes relating to presence or absence of pneumonia and components of gas exchange as well as LOS.

Aims and research question have more consistent outcomes outlined; and lines 152-154 more detail.

Could the explicit aims of IPV be provided so that these could considered with this review of literature and resulting aims; this would also then link to the discussion again later (somewhat addressed in lines 131-136)

Information about IPV use in the literature is presented. The study by Reychler 2018 evaluated airway clearance and gas exchange; was the current review not interested in airway clearance too (albeit in critical care settings)?

Response: Airway clearance is one of the clinical indications for IPV. We did include various search terms to retrieve all the relevant studies that measured “airway clearance” (Please see the supplementary table attached, Table 1S: Search strategy). Despite that, we could not evaluate airway clearance because only three included studies reported that an increased sputum clearance was observed but not measured (quantified) making it vague observation only and therefore difficult to include in our outcome measures; this has been acknowledged and discussed in the Results and Discussion section. 

Please see page 22, Results/Airway clearance: Lines 391-393 “Three studies in this review reported an observed increase in airway clearance with IPV intervention; however, none of them measured the expectorated sputum weight (wet or dry)(12,13,15) or other measures of mucous clearance”.

Please see page 25 & 26, Discussion: Lines 479-484 “Similarly, despite airway clearance being one of the main indications of IPV intervention, only three studies reported an observed increase in secretion clearance. Surprisingly, none of these studies measured sputum yield(12,13,15). Difficulties with accurate measurement of sputum yield have been well documented(32). Due to lack of data regarding the airway clearance in the included studies, the role of IPV in airway clearance remains unclear”

150/151 Care- the role of IPV in preventing or reversing atelectasis is possibly only one only unanswered question.

Response: Reviewer 5’s comments suggest that our sentence (line 150-151) was not clear enough to explain the intention of this review properly. We have therefore rephrased this sentence. 

Please see page 6, Introduction: Lines 155-157 “The question regarding the role of IPV in preventing or reversing atelectasis and reducing the incidence of pneumonia in critically ill patients remains unanswered.”

Line 134: The term conventional chest physiotherapy (ref 12) is used; is there an explanation of this please.

Response: Please see our response to a similar comment by reviewer #1. In response to the reviewer’s question, the term “conventional” has been removed and “chest physiotherapy” is used for consistency throughout the manuscript. 

Please see amendment on page 5, Introduction: Lines 133-135“In addition to these chest physiotherapy (CPT) interventions, intrapulmonary percussive ventilation (IPV) is used in patients with underlying pulmonary…..”

Line 263- is atelectasis one of the outcomes of interest? it seems so

Response: Atelectasis is an outcome of interest. This has been amended and made clearer in the outcome measures section. 

Please see page 12, Methods/Outcome measures: Lines 243-246 “The primary outcome of interest was ICU-LOS. Secondary outcomes included PaO2, the ratio of the partial pressure of arterial oxygen and fraction of inspired oxygen (PaO2/FiO2), PaCO2, airway clearance, the incidence of pneumonia, respiratory rate, and pulmonary atelectasis.”

Lines 264, 265 – this seems in conflict to results presented in the supplementary files

Response: The meta-analysis is performed for LOS and pneumonia outcomes but due to the small number of studies it was not included in the main body of the manuscript, instead all the outcomes are summarised narratively. The meta-analysis, however, still presents some interesting findings which might interest some readers and hence has been included as supplementary material. 

In response to reviewer 5’s comment, we have now changed the description and now meta-analysis is only mentioned in the discussion section for interested readers. 

Please see page 23 & 24, Discussion: Lines 434-437 “A meta-analysis was performed, but due to the small number of studies and observed heterogeneity, it was not included in the main body of this review. Interestingly, pooling of ICU-LOS data revealed that the magnitude and direction of the effect of IPV in reducing the ICU-LOS was similar in all three studies (S1).”

Methods

Patients > or =16years – is this a consistent international cut off for adult critical care units? I am curious that there is potential for studies (included or excluded) conducted in a mixed peadiatric:adult population dependent on the age cut off.

Response: In most Australian hospitals the age group 16 to 17 can be managed in either paediatric or adult unit. To include only 18+ age would potentially exclude a portion of the population that may be managed in adult ICU’s in Australia. 

The mixed study design does detract somewhat from the review and it may be useful to group the presentation of results from the RCTs/quasi randomised and discuss the weight of evidence relating to these.

Additionally, is it possible to appraise and accurately score the quality of an abstract or a non RCT using the PEDro scale which is designed for the quality appraisal of a fully reported RCT(Lines 213 a-215). There are other tools available for appraisal of non RCTs.

Same with Cochrane ROB tool – how can an observational study score under the item “random sequence generation”.

Response: The Cochrane ROB tool is most suited to RCTs, as it assumes that the study being assessed is an RCT. As a result, we considered reviewer’s suggestion and removed three studies from the ROB analysis that did not used randomisation. The new Cochrane ROB table now has 4 studies that are RCTs. Please see Table1 on page 10.

PEDro, however does allow assessment of studies that did not use random allocation; hence we have not made any changes to PEDro assessment table. Due to the scoring system of PEDro tool, it is easy to do head-to-head comparison of all the studies which is much easier for the readers to compare the study quality. Additionally, to our knowledge there is no single assessment tool that would be suitable for the remaining three studies that did not use randomisation as the study design of these three studies that did not use randomisation and their study design differs. This means we may need to use two additional assessment tools which may be confusing and may not help readers with interpretation. 

Results

It may be worth considering the weight/magnitude of the results relating to outcomes from the RCTs; and then presenting the results from the observational studies separately. Otherwise, there is a nice overview of outcome measures included overall (Lines 311-318 onwards), followed by relevant results sections.

Response: Thank you for recognising that we have summarised the study findings nicely.

We have performed meta-analyses (supplementary material “S1 File”) where the direction and magnitude of outcomes are presented for ICU length of stay and pneumonia outcomes. 

Due to small number of studies with each study reporting different outcomes it was difficult to perform this for RCTs and observational studies separately. However as per the suggestion above, we have separated the RCTs from the observational studies in Table 1. Please see table 1 on page 10.

294 presents an overview of the intervention (dose, interface etc). there should be a summary of the control group/comparator groups used.

Response: A description of control group has been added as per the reviewer’s suggestion. 

Please see page 18, Results/Intervention: Lines 296-301 “CPT included chest clapping, postural drainage, expiration with open glottis, incentive spirometer and mobilisation. Duration of CPT session ranged from 30 minutes to 60 minutes once or twice a day. In some studies(15,26), the control group received standard medical treatment which included oxygen therapy, non-invasive ventilation, sitting up in bed, nebulised bronchodilators and corticosteroids (Table 3).”

295/296 – interesting that use of IPV relating to airway clearance is included here, but limited reference to this earlier.

Response: This has been rectified based on similar comments from another reviewer. 

Please see page 12, Methods/Outcome measures: Lines 243-246 “The primary outcome of interest was ICU-LOS. Secondary outcomes included PaO2, the ratio of the partial pressure of arterial oxygen and fraction of inspired oxygen (PaO2/FiO2), PaCO2, airway clearance, the incidence of pneumonia, respiratory rate, and pulmonary atelectasis.” 

Please see page 7, Methods/Inclusion and exclusion criteria: Lines 187-190 “Studies that reported the effects of IPV, high-frequency ventilation, and high-frequency oscillation where these interventions were primarily used intermittently for short duration to promote airway clearance, reverse or treat pulmonary atelectasis”

Lines 361-361- unsure of the term “most significant” here

Response: The term “most significant” has been omitted now. Previously it read “The most significant change in PaO2/FiO2 ratio was reported by Antonaglia….” After amendment it reads “Antonaglia et al. (2006) reported a significant change in PaO2/ FiO2 ratio from admission to discharge….” (Page 20, Results/ PaO2 and PaO2/ FiO2 ratio: Lines 350-352.

Table 1 & 2 – as per methods please consider the suitability of the Cochrane ROB and the PEDro scale for non RCTs.

Response: Thank you for the feedback. We carefully looked into this and found that Cochrane ROB tool is more suited to RCTs as it assumes that the study being assessed is an RCT. As a result, we took the reviewer’s suggestion on board and removed three studies that did not use randomisation. The new Cochrane ROB table now has 4 studies that used random allocation. Please see table 1 on page 10.

PEDro scale, however does allow assessment of studies that did not use random allocation; hence we have not made any changes to PEDro assessment table (Table 2). One advantage of using PEDro tool is that it allows readers to make head-to-head comparisons of all the study qualities directly. Further, to the best of our knowledge there are no assessment tools that would be suitable for the remaining three studies that did not use randomisation as the study design of these three studies do differ. This means we may need to use two additional assessment tools which may detract the readers even more. Please see Table 2.

Discussion

Were there any papers which overlapped with the Reychler 2018 study?

Response: Yes, there were three papers that overlapped with Reychler study.

Line 425 what does routine chest physiotherapy refer to here; please consider the term routine, or use quotations if this is directly from the publication.

Response: The term “Routine chest physiotherapy” has now been clearly explained with terms used by the included studies. “Chest physiotherapy” (CPT) is the used terminology throughout this manuscript for consistency.

Please see Page 22, Discussion: Lines 406-409. “The findings of this review provide weak evidence to support the effectiveness of IPV intervention in reducing ICU-LOS, improving gas exchange and reducing respiratory rate in critically ill patients compared to chest physiotherapy techniques or standard medical management.”

430/431 refer to previous comment in terms of judging ROB and quality.

Response: The ROB and quality has been discussed now in the manuscript. 

(1) Please see page 9, Methods/Assessment of quality and risk of bias: Lines 224-227 “Overall, based on the Cochrane assessment tool, three out of four studies appear to have a low risk of bias whereas in one study(26) the risk of bias was high. On the PEDro scale, the quality of the studies ranged from “poor” to “good”).

(2) Please see Page 27, Limitation: lines 519-520 “Further, small sample sizes and poor methodological quality introduces some bias and weakens the strength of conclusions of this review.”

Also, the assessment table (Table1) has been modified where the observational studies have been removed as per reviewer’s suggestions. Please see Table 1 and Table 2 on page 10 and 11. 

456-459 This is confusing as the review indicated that meta analysis not possible due to heterogeneity. In the supplementary material the I squared statistic does indeed support this high heterogeneity; this implies uncertainty about this specific result and its interpretation (i.e. from the supplementary material).

Response: Meta analyses were not included in the main body of manuscript due to small number of studies and heterogeneities among them, but supplied as supplementary for further reading. The I2 shows heterogeneity but a further statistical analysis such as “p-curve analysis” and “leave one out method” further explains the small effect of heterogeneity on the outcomes. 

We agree with the reviewer that our sentence (“Due to the heterogeneity, meta-analysis could not be included in this review, however, interestingly, pooling of ICU-LOS data revealed that the magnitude …….”) caused some confusion. 

This has been amended, and now it reads “A meta-analysis was performed, but due to the small number of studies and observed heterogeneity, it was not included in the main body of this review. However, interestingly, pooling of ICU-LOS data revealed that the magnitude and direction of the effect of IPV in reducing the ICU-LOS was similar in all three studies (S1).” Please see page 23 & 24, Discussion: Lines 434 and 437.

481-407 Could this discussion be more concise. a/a further pooled data yet this is not reported in the results as according to the methods meta analysis was not feasible.

Line 486 Check the word “don’t”!

Response: We have made the discussion more concise (Please see numerous changes to Discussion section in the tracked version of manuscript supplied).

Further, the reason for not including meta-analysis has been clarified now as above (in the methods and Discussion section). 

Please see page 23 & 24, Discussion: Lines 434 and 437. “A meta-analysis was performed, but due to the small number of studies and observed heterogeneity, it was not included in the main body of this review. However, interestingly, pooling of ICU-LOS data revealed that the magnitude and direction of the effect of IPV in reducing the ICU-LOS was similar in all three studies (S1).” 

Please see page 12, Methods/Outcome measures: Lines 247-248“Due to the small number of studies and observed heterogeneities in the study methodology and patient population, all the outcomes were summarised narratively.”

The word “don’t” has been replaced with word “does not” 

Please see page 25, Discussion: Lines 475-476 “Although statistically significant, this small reduction in pneumonia does not appear to be clinically meaningful”

The discussion is quite long. Perhaps consider a paragraph summarising the clinical implications in terms of patient selection, intervention (timing and dose) etc or add this to the conclusion.

The review sates “more clinical trials with larger sample sizes are warranted to further add to the findings of this review”; this is a very broad statement. Could the authors consider a more focused suggestion to guide the next steps for future research e.g. using the PICO to frame this; It would be interesting to debate what study design should be considered

Response: We have made the discussion more concise (Please see numerous changes to Discussion section in the tracked version of manuscript supplied).

Additionally, as per reviewer’s comments, the authors have created a more focussed recommendation for future research using the PICO framework. 

Please see page 27 and 28, Conclusions: Lines 528-536“This review is based on a small number of available studies mostly with small sample sizes, hence, there is a need for more adequately powered randomised control trials to investigate the effectiveness of IPV intervention in improving outcomes such as ICU LOS, gas exchange, airway clearance, prevention or treatment of pneumonia and pulmonary atelectasis compared to routinely applied airway clearance and lung recruitment physiotherapy interventions in critical care population. In addition, there is also an indication for studies to evaluate patients’ experiences with IPV intervention and their preference compared to routinely practiced respiratory physiotherapy interventions in critical care settings.”

Since this review is based on a small number of studies providing weak evidence, we don’t feel that a recommendation regarding the dosage and duration can be made for clinical implications. However, for readers we have summarised the most common treatment application method and dosage of IPV intervention that was found among the included studies. 

Please see page 18, Results/Intervention: Lines 290-294.“….the duration of a single treatment session ranged from 10 to 30 minutes, and the number of sessions ranged from a single session a day to up to six sessions a day(21). The frequency of delivered breaths remained between 200 to 300 cycles per minute in all the included studies, whereas the airway pressure varied from 5 to 35 cmH2O (Table 3). Notably, most of the studies did not specify the patient's position during the treatment.”

---

## [Decision Letter · Decision Letter 1]

8 Jul 2021

Effect of intrapulmonary percussive ventilation on intensive care unit length of stay, the incidence of pneumonia and gas exchange in critically ill patients: a systematic review

PONE-D-21-11360R1

Dear Dr. Hassan,

We’re pleased to inform you that your manuscript has been judged scientifically suitable for publication and will be formally accepted for publication once it meets all outstanding technical requirements.

Kind regards,

Shane Patman, PhD

Academic Editor

PLOS ONE

Additional Editor Comments (optional):

Reviewers' comments:

Reviewer's Responses to Questions

**Comments to the Author**

1. If the authors have adequately addressed your comments raised in a previous round of review and you feel that this manuscript is now acceptable for publication, you may indicate that here to bypass the “Comments to the Author” section, enter your conflict of interest statement in the “Confidential to Editor” section, and submit your "Accept" recommendation.

Reviewer #1: All comments have been addressed

Reviewer #2: All comments have been addressed

Reviewer #4: All comments have been addressed

Reviewer #5: All comments have been addressed

2. Is the manuscript technically sound, and do the data support the conclusions?

Reviewer #1: Yes

Reviewer #2: Yes

Reviewer #4: Yes

Reviewer #5: (No Response)

3. Has the statistical analysis been performed appropriately and rigorously? 

Reviewer #1: Yes

Reviewer #2: Yes

Reviewer #4: N/A

Reviewer #5: (No Response)

4. Have the authors made all data underlying the findings in their manuscript fully available?

Reviewer #1: Yes

Reviewer #2: Yes

Reviewer #4: Yes

Reviewer #5: (No Response)

5. Is the manuscript presented in an intelligible fashion and written in standard English?

Reviewer #1: Yes

Reviewer #2: Yes

Reviewer #4: Yes

Reviewer #5: (No Response)

6. Review Comments to the Author

Reviewer #1: The comments I previously made have been addressed satisfactorily in the revised manuscript. The authors are to be congratulated for their endeavour to address the comments of all reviewers. The revised manuscript reads clearly and is logically presented.

Reviewer #2: I am satisfied that all queries have been addressed toa satisfactory standard. It is now a strong manuscript.

Reviewer #4: The authors have addressed all of the reviewers major concerns and the resubmission is now acceptable

Reviewer #5: (No Response)

7. PLOS authors have the option to publish the peer review history of their article (what does this mean?). If published, this will include your full peer review and any attached files.

Reviewer #1: **Yes: **Alice Jones

Reviewer #2: No

Reviewer #4: **Yes: **George Ntoumenopoulos

Reviewer #5: No

---

## [Editor Report · Acceptance letter]

15 Jul 2021

PONE-D-21-11360R1 

Effect of intrapulmonary percussive ventilation on intensive care unit length of stay, the incidence of pneumonia and gas exchange in critically ill patients: a systematic review 

Dear Dr. Hassan:

I'm pleased to inform you that your manuscript has been deemed suitable for publication in PLOS ONE. Congratulations! Your manuscript is now with our production department. 

Kind regards, 

on behalf of

Assoc Prof Shane Patman 

Academic Editor

PLOS ONE